# Learning from Distributed Users in Contextual Linear Bandits Without Sharing the Context

**Osama A. Hanna**
University of California, Los Angeles
ohanna@ucla.edu

**Lin F. Yang**
University of California, Los Angeles
linyang@ucla.edu

**Christina Fragouli**
University of California, Los Angeles
christina.fragouli@ucla.edu

## Abstract

Contextual linear bandits is a rich and theoretically important model that has many practical applications. Recently, this setup gained a lot of interest in applications over wireless where communication constraints can be a performance bottleneck, especially when the contexts come from a large $d$-dimensional space. In this paper, we consider a distributed memoryless contextual linear bandit learning problem, where the agents who observe the contexts and take actions are geographically separated from the learner who performs the learning while not seeing the contexts. We assume that contexts are generated from a distribution and propose a method that uses $\approx 5d$ bits per context for the case of unknown context distribution and $0$ bits per context if the context distribution is known, while achieving nearly the same regret bound as if the contexts were directly observable. The former bound improves upon existing bounds by a $\log(T)$ factor, where $T$ is the length of the horizon, while the latter achieves information theoretical tightness.

## 1 Introduction

Contextual linear bandits offer a sequential decision-making framework that combines fundamental theoretical importance with significant practical popularity [8], as it offers a tractable way to capture side information (context), as well as a potentially infinite set of decisions (actions). The most prominent application is in recommendation systems [30], but it has also been used in applications such as virtual support agents [39], clinical trials [12], transportation systems [9], wireless optimization [26, 25], health [10], robotics [31] and online education [34].

In this paper, we develop algorithms that support the deployment of contextual linear bandits in distributed settings. In particular, we consider the case where a central learner wishes to solve a contextual linear bandit problem with the help of transient agents. That is, we assume that the agents do not keep memory of past actions and may not be present for the whole duration of learning; learning in our setup can happen thanks to the persistent presence of the central learner. We view the central learner as a "knowledge repository", that accumulates knowledge from the experience of the transient agents and makes it available to next agents. The central learner, through the information it keeps, could help passing by devices decide how to perform an action, for example: passing by drones decide how to perform a manoeuver; agricultural robots decide what amounts of substances such as pesticids to release; and passing by mobile devices decide which local restaurants to recommend.

The main challenge we try to address in this paper is the efficient communication of the context the agents experience. More specifically, in our setup, each time an agent joins, she receives from the

36th Conference on Neural Information Processing Systems (NeurIPS 2022).

central learner information on the system, such as current estimates of the system parameters; she observes her current context, selects and plays an action and collects the corresponding reward. Note that although the distributed agent knows her context, the action she plays and the observed reward, the central learner does not - and needs this information to update its estimate of the system parameters. The context in particular can be communication heavy - in the examples we mentioned before, for drones the context could be their navigation capabilities, physical attributes, and enviromental factors such as wind speed; for agricultural robots, it could be images that indicate state of plants and sensor measurements such as of soil consistency; for restaurant recommendations, it could be the personal dietary preferences and restrictions, budget, and emotional state. Moreover, because of geographical separation, the central learner may not have any other way to learn the context beyond communication. Unlike the reward, that is usually a single scalar value, the context can be a vector of a large dimension $d$ from an infinite alphabet, and thus, communicating the context efficiently is heavily nontrivial.

The technical question we ask is, *how many bits do we need to convey per context to solve the linear bandit problem without downgrading the performance as compared to the non-distributed setting?*

In this paper, we design algorithms that support this goal. We note that our algorithms optimize the uplink communication (from the agents to the central learner), and assume unlimited (cost-free) downlink communication. This is a standard assumption in wireless [7, 33, 21] for several reasons: uplink wireless links tend to be much more bandwidth restricted, since several users may be sharing the same channel; uplink communication may also be battery-powered and thus more expensive to sustain; in our particular case, the agents may have less incentive to communicate (provide their feedback) than the central learner (who needs to learn). Having said that, we note that our algorithms (in Sections 3 and 4) make frugal use of the downlink channels, only using them to transmit system parameters.

Below we summarize our main contributions:
**1.** We show the surprising result that, if the central learner knows the distribution of the contexts, we do not need to communicate the context at all - the agent does not need to send any information on the actual context she observes and the action she plays. It is sufficient for the agent to just send $1$ bit to convey quantized information on her observed reward and nothing else. But for this very limited communication, the central learner can learn a policy that achieves the same order of regret as if full information about the context and reward is received. This result holds for nearly all context distributions and it is the best we can hope for - *zero bits* of communication for the context.
**2.** If the central learner has no knowledge of the context distribution, we show that $\approx 5d$ bits per context (where $d$ is the context dimension) is sufficient to achieve the same order regret as knowing the context in full precision. Note that previous algorithms, that rely on constructing $1/T$-net for the set of feature vectors, use $O(d \log T)$ bits per context to achieve the same order regret, where $T$ is the length of the horizon [24], and require time complexity of $O(T^d)$ which is exponential in $d$.

**Related Work and Distinction.** Contextual linear bandits is a rich and important model that has attracted significant interest both in theory and applications [8, 24]. Popular algorithms for this setup include LinUCB [1, 37] and cotextual Thompson sampling [2]. Under Assumption 1, these algorithms achieve a regret of $\tilde{O}(d\sqrt{T})$, where $d$ is the dimension of an unknown system parameter and $T$ is the time horizon, while the best known lower bound for this setup is $\Omega(d\sqrt{T})$ [37]. These algorithms assume perfect knowledge of the contexts and rewards. Within this space, our work focuses on operation under communications constraints in a distributed setting.

There is large body of work focusing on distributed linear contextual bandits settings, but mainly within the framework of federated learning, where batched algorithms have been proposed for communication efficiency [43, 41, 6, 5, 23] that aggregate together observations and parameter learning across a large number of iterations. This is possible because in federated learning, the agents themselves wish to learn the system parameters, remain active playing multiple actions throughout the learning process, and exchange information with the goal of speeding up their learning [43, 41]. In contrast, in our setup batched algorithms cannot reduce the communication cost because each agent only plays a single action; this may be because agents are transient, but also because they may not be interested in learning - this may not be a task that the agents wish to consistently perform - and thus do not wish to devote resources to it. For example, an agent may wish to try a restaurant in a special occasion, but would not be interested in sampling multiple restaurants/learning recommendation system parameters. In other words, we consider a scenario where the user benefits from receiving an action (or policy) from the central learner, e.g., a recommendation. In response, the user gives

feedback to the central learner in terms of (compressed) context/reward. The compression operations benefit the user by helping reduce her communication cost. In principle, the user is not required to respond. But the central learner will be able to learn whenever there is a feedback; creating an incentive for the user to respond could be an interesting future topic. Our setup supports a different (and complementary) set of applications than the federated learning framework, and requires a new set of algorithms that operate without requiring agents to keep memory of past actions.[1]

There is a long line of research on compression for machine learning and distributed optimization, e.g., compression for distributed gradient descent [40, 3, 32, 18], and distributed inference [19]. However, such schemes are not optimized for active learning applications. Our compression schemes can be seen as quantization schemes for contexts and rewards tailored to active learning applications.

Our work also differs from traditional vector compression schemes [15] that aim to reconstruct the data potentially with some distortion (achieve rate-distortion trade-offs). In our case, we do not aim to reconstruct the data, but instead to distinguish the best arm for each context. Indeed, using 0 bits, as we do in Section 3, we cannot reconstruct a meaningful estimate of the context.

To the best of our knowledge, our framework has not been examined before for linear contextual bandits. Work in the literature has examined compression for distributed memoryless MABs [21], but only for rewards (scalar values) and not the contexts (large vectors), and thus these techniques also do not extend to our case.

**Paper organization.** Section 2 reviews our notation and problem formulation; Section 3 provides and analyzes our algorithm for known and Section 4 for unknown context distributions.

## 2 Notation and Problem Formulation

**Notation.** We use the following notation throughout the paper. For a vector $X$ we use $X_i$ or $(X)_i$ to denote the $i$-th element of the vector $X$; similarly for a matrix $V$, we use $V_{ij}$ or $(V)_{ij}$ to denote the element at row $i$, and column $j$. We use $\|V\|_2$ to denote the matrix spectral norm. For a function $f$, we denote its domain and range by $\text{dom}(f)$, $\text{ran}(f)$ respectively. When $\text{dom}(f) \subseteq \mathbb{R}$, we use $f(X)$ for a vector $X \in \mathbb{R}^d$ to denote $f(X) := [f(X_1), ..., f(X_d)]$, i.e., the function $f$ is applied element-wise; for example we use $X^2$ to denote the element-wise square of $X$. We denote the inverse of a function $f$ by $f^{-1}$; if $f$ is not one-to-one, with abuse of notation we use $f^{-1}$ to denote a function that satisfies $f(f^{-1}(x)) = x \forall x \in \text{ran}(f)$ (this is justified due to the axiom of choice [22]). For a matrix $V$, we use $V^{-1}$ to denote its inverse; if $V$ is singular, we use $V^{-1}$ to denote its pseudo-inverse. We use $[N]$ for $N \in \mathbb{N}$ to denote $\{1, ..., N\}$, and $\{X_a\}_{a \in \mathcal{A}}$ to denote the set $\{(a, X_a) | a \in \mathcal{A}\}$. We say that $y = O(f(x))$ if there is $x_0$ and a constant $C$ such that $y \leq Cf(x) \; \forall x > x_0$; we also use $\tilde{O}(f(x))$ to omit log factors.

**Contextual Linear Bandits.** We consider a contextual linear bandits problem over a horizon of length $T$ [8], where at each iteration $t = 1, ..., T$, an agent, taking into account the context, chooses an action $a_t \in \mathcal{A}$ and receives a reward $r_t$. For each action $a \in \mathcal{A}$, the agent has access to a corresponding feature vector $X_{t,a} \in \mathbb{R}^d$. The set of all such vectors $\{X_{t,a}\}_{a \in \mathcal{A}}$ is the context at time $t$, and the agent can use it to decide which action $a_t$ to play. We assume that the context is generated from a distribution, i.e., given $a$, $X_{t,a}$ is generated from a distribution $\mathcal{P}_a$. As a specific example, we could have that $a \in \mathbb{R}^d$ and $X_{t,a}$ is generated from a Gaussian distribution with zero mean and covariance matrix $\|a\|_2 I$, where $I$ is the identity matrix, i.e., $\mathcal{P}_a = \mathcal{N}(0, \|a\|_2 I)$. The selection of $a_t$ may depend not only on the current context $\{X_{t,a}\}_{a \in \mathcal{A}}$ but also on the history $H_t \triangleq \{\{X_{1,a}\}_{a \in \mathcal{A}}, a_1, r_1, ..., \{X_{t-1,a}\}_{a \in \mathcal{A}}, a_{t-1}, r_{t-1}\}$, namely, all previously selected actions, observed contexts and rewards. Once an action is selected, the reward is generated according to

$$r_t = \langle X_{t,a_t}, \theta_\star \rangle + \eta_t, \tag{1}$$

where $\langle ., . \rangle$ denotes the dot product, $\theta_\star$ is an unknown (but fixed) parameter vector in $\mathbb{R}^d$, and $\eta_t$ is noise. We assume that the noise follows an unknown distribution with $\mathbb{E}[\eta_t | \mathcal{F}_t] = 0$ and $\mathbb{E}[\exp(\lambda \eta_t) | \mathcal{F}_t] \leq \exp(\lambda^2/2) \forall \lambda \in \mathbb{R}$, where $\mathcal{F}_t = \sigma(\{X_{1,a}\}_{a \in \mathcal{A}}, a_1, r_1, ..., \{X_{t,a}\}_{a \in \mathcal{A}}, a_t)$ is the filtration [13] of historic information up to time $t$, and $\sigma(X)$ is the $\sigma$-algebra generated by $X$ [13].

---

[1]Our techniques could be adapted to additionally improve the communication efficiency of batched algorithms, but this is not the focus of our work.

The objective is to minimize the regret $R_T$ over a horizon of length $T$, where

$$R_T = \sum_{t=1}^{T} \max_{a \in \mathcal{A}} \langle X_{t,a}, \theta_\star \rangle - \langle X_{t,a_t}, \theta_\star \rangle. \tag{2}$$

The best performing algorithms for this problem, such as LinUCB and contextual Thompson sampling, achieve a worst case regret of $\tilde{O}(d\sqrt{T})$ [29, 28, 1, 2]. The best known lower bound is $\Omega(d\sqrt{T})$ [37].

In the rest of this paper, we make the following assumptions that are standard in the literature [24].

**Assumption 1.** *We consider contextual linear bandits that satisfy:*
**(1.)** $\|X_{t,a}\|_2 \le 1$, $\forall t \in [T]$, $a \in \mathcal{A}$.     **(2.)** $\|\theta_\star\|_2 \le 1$.     **(3.)** $r_t \in [0,1]$, $\forall t \in [T]$.

The boundedness assumption on $r_t$ can be relaxed using [21], which only requires approximately 3.5 bits on average to send $r_t$, even if it is unbounded.

**Memoryless Distributed Contextual Linear Bandits.** We consider a distributed setting that consists of a central learner communicating with geographically separated agents. For example, the agents are drones that interact with a traffic policeman (central learner) as they fly by. We assume that the agents do not keep memory of past actions and may not be present for the whole duration of learning; learning in our setup can happen thanks to the persistent presence of the central learner.

At each time $t$, $t = 1 \ldots T$, a distributed agent joins the system; she receives from the central learner information on the system, such as the current estimate of the parameter vector $\theta_\star$ or the history $H_t$; she observes the current context $\{X_{t,a}\}_{a \in \mathcal{A}}$, selects and plays an action $a_t$ and collects the corresponding reward $r_t$. Note that although the distributed agent knows the context $\{X_{t,a}\}_{a \in \mathcal{A}}$, the action $a_t$ and the observed reward $r_t$, the central learner does not. The central learner may need this information to update its estimate of the system parameters, such as the unknown parameter vector $\theta_*$, and the history $H_{t+1}$. However, we assume that the agent is restricted to utilize a communication-constrained channel and thus may not be able to send the full information to the central learner.

The main question we ask in this paper is: can we design a compression scheme, where the agent sends to the central learner only one message using $B_t$ bits (for as small as possible a value of $B_t$) that enables the central learner to learn equally well (experience the same order of regret) as if there were no communication constraints? With no communication constraints the agent could send unquantized the full information $\{\{X_{t,a}\}_{a \in \mathcal{A}}, a_t, r_t\}$. Instead, the agent transmits a message that could be a function of all locally available information at the agent. For example, it could be a function of $(H_t, \{X_{t,a_t}\}_{a \in \mathcal{A}}, a_t, r_t)$, if the agent had received $H_t$ from the central learner. It could also be a function of just $(X_{t,a_t}, r_t)$, which could be sufficient if the central learner employs an algorithm such as LinUCB [1, 37]. In summary, we set the following goal.

**Goal.** Design contextual linear bandit schemes for the memoryless distributed setting that achieve the best known regret of $O(d\sqrt{T\log(T)})$, while communicating a small number of bits $B_t$.

We only impose communication constraints on the uplink communication (from the agents to the central learner) and assume no cost downlink communication (see discussion in Secttion 1).

**Stochastic Quantizer (SQ) [16].** Our proposed algorithms use stochastic quantization, that we next review. We define $\text{SQ}_\ell, \ell \in \mathbb{N}$ to be a quantizer, that uses $\log(\ell+1)$ bits, consisting of an encoder and decoder described as following. The encoder $\xi_\ell$ takes a value $x \in [0, \ell]$ and outputs an integer value

$$\xi_\ell = \begin{cases} \lfloor x \rfloor & \text{with probability } \lceil x \rceil - x \\ \lceil x \rceil & \text{with probability } x - \lfloor x \rfloor. \end{cases} \tag{3}$$

The output $\xi_\ell$ is represented with $\log(\ell+1)$ bits. The decoder $D_\ell$ takes as input the binary representation of $\xi_\ell(x)$ and outputs the real value $\xi_\ell(x)$. The composition of the encoder $\xi_\ell(x)$, the binary mapping, and decoder $D_\ell$ is denoted by $\text{SQ}_\ell$. We notice that since the decoder only inverts the binary mapping operation, we have that $\text{SQ}_\ell = \xi_\ell$. When $\text{SQ}_\ell$ is applied at the agents side, the agent encodes its data, $x$, as $\xi_\ell(x)$, then sends the corresponding binary mapping to the central learner that applies $D_\ell$ to get $\text{SQ}_\ell(x)$. With slightly abuse of notation, this operation is described in the paper, by saying that the agent sends $\text{SQ}_\ell$ to the central learner.

The quantizer $\text{SQ}_\ell$ is a form of dithering [16] and it has the following properties

$$\mathbb{E}[\text{SQ}_\ell(x)|x] = \lfloor x \rfloor (\lceil x \rceil - x) + \lceil x \rceil (x - \lfloor x \rfloor) = x(\lceil x \rceil - \lfloor x \rfloor) = x, \quad \text{and} \quad |\text{SQ}_\ell(x) - x| \le 1.$$

In particular, it conveys an unbiased estimate of the input with a difference that is bounded by $1$ almost surely. We also define a generalization of $\text{SQ}_\ell$ denoted by $\text{SQ}_\ell^{[a,b]}$ where the input $x$ of the encoder is in $[a,b]$ instead of $[0,\ell]$. The encoder first shifts and scales $x$ using $\tilde{x} = \frac{\ell}{b-a}(x-a)$ to make it lie in $[0,\ell]$, then feeds $\tilde{x}$ to the encoder in (3). This operation is inverted at the decoder. It is easy to see that $\text{SQ}_\ell^{[a,b]}$ satisfies

$$\mathbb{E}[\text{SQ}_\ell^{[a,b]}(x)|x] = x, |\text{SQ}_\ell^{[a,b]}(x) - x| \le \frac{b-a}{\ell}.$$

## 3    Contextual Linear Bandits with Known Context Distribution

In this section, we show that if the central learner knows the distributions for the vectors $X_{t,a}$, then the agent does not need to convey the specific realization of the vector $X_{t,a}$ she observes at all - it is sufficient to just send 1 bit to convey some information on the observed reward and nothing else. But for this very limited communication, the central learner can experience the same order regret, as when receiving in full precision all the information that the agents have, namely, $R_T = O(d\sqrt{T \log T})$. Algorithm 1, that we describe in this section, provides a method to achieve this. Algorithm 1 is clearly optimal, as we cannot hope to use less than zero bits for the vector $X_{t,a}$.

**Remark 1.** Knowledge of the distribution of $X_{t,a}$ is possible in practice, since many times the context may be capturing well studied statistics (e.g., male or female, age, weight, income, race, dietary restrictions, emotional state, etc) - the advent of large data has made and will continue to make such distributions available. Similarly, actions may be finite (eg., restaurants to visit) or well described (e.g., released amounts of substances), and thus the distribution of $X_{t,a}$ could be derived. When the distribution is approximately known, we provide later in this section a bound on the misspefication performance penalty in terms of regret.

**Main Idea.** The intuition behind Algorithm 1 is that it reduces the multi-context linear bandit problem to a single context problem. In particular, it calls as a subroutine an algorithm we term $\Lambda$, that serves as a placeholder for any current (or future) bandit algorithm that achieves regret $O(d\sqrt{T \log T})$ for the case of a single context (for example, LinUCB [1, 37]). The central learner uses $\Lambda$ to convey to the agents the information they need to select a good action. Our aim is to parametrize the single context problem appropriately, so that, by solving it we also solve our original problem.

Recall that in a single context problem, at each iteration $t$, any standard linear bandit algorithm $\Lambda$ selects a feature vector (an action) $x_t$ from a set of allowable actions $\mathcal{X}$, and observes a reward

$$r_t = \langle x_t, \theta'_\star \rangle + \eta_t, \tag{4}$$

where $\theta'_\star$ is an uknown parameter and $\eta_t$ is noise that satisfies the same assumptions as in (1). The objective of $\Lambda$ is to minimize the standard linear regret $R_T(\Lambda)$ over a horizon of length $T$, namely

$$R_T(\Lambda) = \sum_{t=1}^{T} \max_{x \in \mathcal{X}} \langle x, \theta'_\star \rangle - \langle x_t, \theta'_\star \rangle. \tag{5}$$

Our reduction proceeds as follows. We assume that $\Lambda$ operates over the same horizon of length $T$ and is parametrized by an unknown parameter $\theta'_\star$. We will design the action set $\mathcal{X}$ that we provide to $\Lambda$ using our knowledge of the distributions $\mathcal{P}_a$[2] as we will describe later in (7). During each iteration, the central learner asks $\Lambda$ to select an action $x_t \in \mathcal{X}$ and then provides to $\Lambda$ a reward for this action (our design ensures that this reward satisfies (4) with $\theta'_\star = \theta_\star$). $\Lambda$ operates with this information, oblivious to what else the central learner does. Yet, the action $x_t$ is never actually played: the central learner uses the selected action $x_t$ to create an updated estimate of the parameter vector $\hat{\theta}_t$, as we will describe later, and only sends this parameter vector estimate to the distributed agent. The agent observes her context, selects what action to play, and sends back her observed quantized reward to the central learner. This is the reward that the central learner provides to $\Lambda$. We design the set $\mathcal{X}$ and the agent operation to satisfy that: (4) holds; and $R_T - R_T(\Lambda)$ is small, where $R_T$ is the regret for our original multi-context problem and $R_T(\Lambda)$ the regret of $\Lambda$. We next try to provide some intuition on how we achieve this.

We first describe how we construct the set $\mathcal{X}$. Let $\Theta$ be the set of all values that $\theta_\star$ could possibly take. For each possible parameter vector value $\theta \in \Theta$ the central learner considers the quantity

$$X^\star(\theta) = \mathbb{E}_{\{x_a : x_a \sim P_a\}}[\arg \max_{x \in \{x_a : a \in A\}} \langle x, \theta \rangle] \tag{6}$$

---

[2]Recall that given $a$, $X_{t,a}$ is generated from distribution $\mathcal{P}_a$, see Section 2.

where $x_a$ is the random variable that follows the distribution $\mathcal{P}_a$. Ties in (6) can be broken uniformly at random. In fact any pre-selected choice function would work as long as the same function is also used in step 12 of Algorithm 1. Note that the function $X^\star : \mathbb{R}^d \to \mathbb{R}^d$ can be computed offline before the learning starts, see Example 1. We then use

$$\mathcal{X} = \{X^\star(\theta) | \theta \in \Theta\}. \tag{7}$$

Intuitively, for each value of $\theta$, we optimistically assume that the distributed agent may select the best possible realization $X_{t,a}$ for this $\theta$ (that has the expectation in (6)), and receive the associated reward; accordingly, we restrict the action space $\mathcal{X}$ of $\Lambda$ to only contain the expectation of these "best" $X_{t,a}$. The vector $x_t \in \mathcal{X}$ may not actually be the vector corresponding to the action the agent selects; it is only used to convey to the agent an estimate of the unknown parameter $\hat{\theta}_t$ that satisfies $x_t = X^\star(\hat{\theta}_t)$. Although the central learner does not control which action the agent plays, this is influenced by $\hat{\theta}_t$; we show in App. A that $X_{t,a_t}$ is an unbiased estimate of $x_t$, and the generated reward follows the linear model in (4) with $\theta'_\star = \theta_\star$. In Theorem 1, we prove that

$$\arg\max_{x \in \mathcal{X}} \langle x, \theta_\star \rangle = X^\star(\theta_\star). \tag{8}$$

Hence, if $\Lambda$ converges to selecting the best action for the single context problem, we will have that $\hat{\theta}_t$ converges to $\theta_\star$ if the maximizer in (8) is unique. If there are multiple values for $\theta$ with $X^\star(\theta) = X^\star(\theta_\star)$, we show in the proof of Theorem 1 that they all lead to the same expected reward for the original multi-context problem.

**Example 1.** Consider the case where $d = 1$, $\mathcal{A} = \{1, 2\}$, $X_{t,a} \in \{-1, 1\} \ \forall a \in \mathcal{A}$, $\Theta = \{-1, 1\}$, $\theta_\star = 1$ and $X_{t,1}$ takes the value $-1$ with probability $p$ and $1$ otherwise, while $X_{t,2}$ takes the values $-1$ with probability $q$ and $1$ otherwise. Then, we have that

$$\arg\max_{X_{t,a}} \langle X_{t,a}, 1 \rangle = \begin{cases} 1 & \text{with probability } 1 - pq \\ -1 & \text{with probability } pq, \end{cases} \tag{9}$$

where we use the fact that if $\arg\max_{X_{t,a}} \langle X_{t,a}, 1 \rangle \neq 1$, it must be the case that both $X_{t,1}$ and $X_{t,2}$ are $-1$. Thus, $X^\star(1) = \mathbb{E}[\arg\max_{X_{t,a}} \langle X_{t,a}, 1 \rangle] = 1 - 2pq$, and similarly $X^\star(-1) = -1 + 2(1-p)(1-q)$, and hence, $\mathcal{X} = \{1 - 2pq, -1 + 2(1-p)(1-q)\}$. If $\Lambda$ decides to pick $x_t = 1 - 2pq$, we have that $\hat{\theta}_t = 1$, otherwise $\hat{\theta}_t = -1$. This estimate $\hat{\theta}_t$ is then conveyed to the agent to help her pick the action.

**Algorithm Operation.** The pseudo-code is provided in Algorithm 1.
• First, the central learner calculates the function

$$X^\star(\theta) = \mathbb{E}_{\{x_a : x_a \sim P_a\}}[\arg\max_{x \in \{x_a : a \in A\}} \langle x, \theta \rangle], \tag{10}$$

and creates the action set $\mathcal{X} = \{X^\star(\theta) | \theta \in \Theta\}$ that algorithm $\Lambda$ is going to use.
• At each time $t$, based on past history, $\Lambda$ decides on a next action $x_t \in \mathcal{X}$. The central learner uses $x_t$ to calculate the new update $\hat{\theta}_t = X^{-1}(x_t)$, where $X^{-1}$ is the inverse of $X^\star$ (see Section 2).
• The agent receives $\hat{\theta}_t$ from the central learner, observes her context, plays an action $a_t = \arg\max_{a \in \mathcal{A}} \langle X_{t,a}, \hat{\theta}_t \rangle$, and observes the reward $r_t$. She then quantizes the reward using a stochastic quantizer $SQ_1$ (see Section 2), and communicates the outcome using one bit to the central learner.
• The central learner provides the quantized reward as input to $\Lambda$. Note that $\Lambda$ is oblivious to what actions are actually played; it treats the received reward as corresponding to the action $x_t$ it had decided.

The following theorem proves that Algorithm 1 achieves a regret $R_T(\Lambda) + O(\sqrt{T \log T})$, where $R_T(\Lambda)$ is the regret of $\Lambda$ in (5). Hence, if $\Lambda$ satisfies the best known regret bound of $O(d\sqrt{T \log T})$, e.g., LinUCB, Algorithm 1 achieves a regret of $O(d\sqrt{T \log T})$. The theorem holds under the mild set of assumptions that we stated in Section 2.

**Theorem 1.** *Algorithm 1 uses* 1 *bit per reward and* 0 *bits per context. Under Assumption 1, it achieves a regret* $R_T = R_T(\Lambda) + O(\sqrt{T \log T})$ *with probability at least* $1 - \frac{1}{T}$.

**Proof outline.** The complete proof is available in App. A. We next provide a short outline. From the definition of $X^\star$ in (10), we notice the following. Recall that the distributed agent receives $\hat{\theta}_t$ from the central learner, and pulls the best action for this $\hat{\theta}_t$, i.e., $a_t = \arg\max_{a \in \mathcal{A}} \langle X_{t,a}, \hat{\theta}_t \rangle$. We

---

**Algorithm 1** Communication efficient for contextual linear bandits with known distribution

---
1: Input: an algorithm $\Lambda$ for one context case, underlying set of actions $\mathcal{X}$, and time horizon $T$.
2: Initialize: $X^\star(\theta) = \mathbb{E}_{\{x_a : x_a \sim P_a\}}[\arg\max_{x \in \{x_a : a \in A\}} \langle x, \theta \rangle]$, $\mathcal{X} = \{X^\star(\theta) | \theta \in \Theta\}$, $\hat{r}_0 = 0$.
3: Let $X^{-1}$ be an inverse of $X^\star$.
4: **for** $t = 1 : T$ **do**
5:     **Central learner**:
6:         Receive $\hat{r}_{t-1}$ and provide it to $\Lambda$.
7:         $\Lambda$, using the history $(x_1, \hat{r}_1, ..., x_{t-1}, \hat{r}_{t-1})$, selects $x_t$.
8:         Send $\hat{\theta}_t = X^{-1}(x_t)$ to agent.
9:     **Agent**:
10:        Receive $\hat{\theta}_t$ from the central learner.
11:        Observe context realization $\{X_{t,a}\}_{a \in \mathcal{A}}$.
12:        Pull arm $a_t = \arg\max_{a \in \mathcal{A}} \langle X_{t,a}, \hat{\theta}_t \rangle$ and receive reward $r_t$.
13:        Send $\hat{r}_t = \mathrm{SQ}_1(r_t)$ to the central learner using 1-bit.

---

show that conditioned on $x_t$, the associated vector $X_{t,a_t}$ is an unbiased estimate of $x_t$ with a small variance. Given this, we prove that $\hat{r}_t$ satisfies (6), and thus the rewards observed by $\Lambda$ are generated according to a linear bandit model with unknown parameter that is the same as $\theta_\star$.

We next decompose the difference $R_T - R_T(\Lambda)$ to two terms: $\Sigma_T = \sum_{t=1}^{T} \langle \arg\max_{X_{t,a}} \langle X_{t,a}, \theta_\star \rangle, \theta_\star \rangle - \langle x_t, \theta_\star \rangle$ and $\Sigma'_T = \sum_{t=1}^{T} \langle \arg\max_{X_{t,a}} \langle X_{t,a}, \hat{\theta}_t \rangle, \theta_\star \rangle - \max_{x \in \mathcal{X}} \langle x, \theta_\star \rangle$. To bound the first term, we show that the unbiasdness property together with Assumption 1 implies that $\Sigma_T$ is a martingale with bounded difference. This implies that $|\Sigma_T| = O(\sqrt{T \log T})$ with high probability. To bound $\Sigma'_T$, we first show that $\arg\max_{x \in \mathcal{X}} \langle x, \theta_\star \rangle = X^\star(\theta_\star)$ (we note that this is why the algorithm converges to $\hat{\theta}_t$ that is equal to, or results in the same expected reward as, $\theta_\star$). Then, following a similar approach, we can show that $\Sigma'_T$ is a martingale with bounded difference which implies that $|\Sigma'_T| = O(\sqrt{T \log T})$ with high probability. □

**Downlink Communication.** The downlink cost of our scheme is $O(d)$ (see App. A for discussion).

**Operation Complexity.** The main complexity that our algorithm adds beyond the complexity of $\Lambda$, is the computation of the function $X^\star$. The time-complexity of $X^*(\theta)$ depends on the context distribution. While computing $X^*(\theta)$ can be computationally expensive in worst-case scenarios, it can be computed/approximated efficiently for many practical distributions even in a closed form. We give the following examples:

• For $d = 1, \theta > 0$, we have that $X^\star(\theta)$ is the expectation of the maximum of multiple random variables, i.e., $X^\star(\theta) = \mathbb{E}_{x_a \sim \mathcal{P}_a}[\max_{a \in \mathcal{A}} x_a]$, which can be computed/approximated efficiently if the distributions $\mathcal{P}_a$ are given in a closed form.

• If $\{P_a\}_{a \in \mathcal{A}}$ are continuous distributions, then $X^*(\theta), \theta \neq 0$ can be expressed as

$$X^*(\theta) = \sum_{a \in \mathcal{A}} \int_{x_a \sim P_a} x_a E_{x_{a'} \sim P_{a'}, a' \in \mathcal{A}/\{a\}}[I[\langle x_{a'}, \theta \rangle < \langle x_a, \theta \rangle \forall a' \neq a]|x_a]dP_a. \quad (11)$$

For many distributions, the previous expression can be computed/approximated efficiently. For instance, consider the case where $d \geq 1$, $x_a$ are independent, identically distributed $d$-dimensional Gaussian vectors with mean $\mu$ and covariance matrix $\Sigma = U^T DU$, where $D$ is a diagonal matrix and $U$ is upper triangular. The expectation in (11) is equal to $(Q(\frac{\langle x_a - \mu, \theta \rangle}{\|\sqrt{D}U\theta\|_2}))^{|\mathcal{A}|-1}$, where $Q(c) = \frac{1}{\sqrt{2\pi}} \int_c^\infty \exp(-\frac{1}{2}x^2)dx$. Hence, $X^*(\theta)$ can be approximated efficiently in that case.

• For discrete distributions, $X^*(\theta)$ can be computed efficiently depending on the number of mass points of the distribution and if the distribution has structures/properties to simplify the expression.

**Imperfect Knowledge of Distributions.** Since we only use the distributions to calculate $X^\star$, imperfect knowledge of distribution only affects us in the degree that it affects the calculation of $X^\star$. Suppose that we have an estimate $\tilde{X}^\star$ of $X^\star$ that satisfies

$$\sup_{\theta \in \Theta} \|X^\star(\theta) - \tilde{X}^\star(\theta)\|_2 \leq \epsilon. \quad (12)$$

Using Theorem 1 we prove in App. A the following corollary.

**Corollary 1.** Suppose we are given $\tilde{X}^\star$ that satisfies (12). Then, there exists an algorithm $\Lambda$ for which Algorithm 1 achieves $R_T = \tilde{O}(d\sqrt{T} + \epsilon T\sqrt{d})$ with probability at least $1 - \frac{1}{T}$.

**Privacy.** Our result may be useful for applications beyond communication efficiency; indeed, the context may contain private information (e.g., personal preferences, financial information, etc); use of our algorithm enables to not share this private information at all with the central learner, without impeding the learning process. Surprisingly, work in [48], motivated from privacy considerations, has shown that if an agent adds a small amount of zero mean noise to the true context before sending it to the central learner, this can severely affect the regret in some cases - and yet our algorithm essentially enables to "guess" the context with no regret penalty if the distributions are known. Although adding a zero mean noise to the observed feature vector conveys an unbiased estimate of the observation, the difference between this and our case is technical and mainly due to the fact that the unbiasdness is required to hold conditioned on the central learner observation (noisy context).

Note that we do not make formal privacy claims in this paper, but simply observe that our approach could potentially be leveraged for privacy purposes. It is true that the reward can reveal some information about the context, e.g., if all the actions result in small reward for context and large reward for another context. However, privatizing the reward (which implies a private context in our case) is much easier than privatizing the context and there are many proposed optimal algorithms with little to no regret loss, e.g., see [20, 38, 44, 35]. This is not the case when privatizing the context. In fact it was shown in [42] that privatizing the context can lead to linear regret and relaxed definitions of privacy are proposed to avoid this.

## 4 Contextual Linear Bandits with Unknown Context Distribution

We now consider the case where the learner does not know the context distributions, and thus Algorithm 1 that uses zero bits for the context cannot be applied. In this case, related literature conjectures a lower bound of $\Omega(d)$ [46, 47] – which is discouraging since it is probably impossible to establish an algorithm with communication logarithmically depending on $d$. Additionally, in practice we use $32d$ bits to convey full precision values - thus this conjecture indicates that in practice we may not be able to achieve order improvements in terms of bits communicated, without performance loss.

In this section, we provide Algorithm 2 that uses $\approx 5d$ bits per context and achieves (optimal) regret $R_T = O(d\sqrt{T\log T})$. We believe Algorithm 2 is interesting for two reasons:
1. In theory, we need an infinite number of bits to convey full precision values- we prove that a constant number of bits per dimension per context is sufficient. Previously best-known algorithms, which rely on constructing $1/T$-net for the set of feature vectors, use $O(d\log T)$ bits per context, which goes to infinity as $T$ goes to infinity. Moreover, these algorithms require exponential complexity [24] while ours is computationally efficient.
2. In practice, especially for large values of $d$, reducing the number of bits conveyed from $32d$ to $\approx 5d$ is quite significant - this is a reduction by a factor of six, which implies six times less communication.

**Main Idea.** The intuition behind Algorithm 2 is the following. The central learner is going to use an estimate of the $d \times d$ least-squares matrix $V_t = \sum_{i=1}^{t} X_{i,a_i} X_{i,a_i}^T$ to update her estimates for the parameter vector $\theta_\star$. Thus, when quantizing the vector $\bar{X}_{t,a}$, we want to make sure that not only this vector is conveyed with sufficient accuracy, but also that the central learner can calculate the matrix $V_t$ accurately. In particular, we would like the central learner to be able to calculate an unbiased estimator for each entry of $X_{t,a}$ and each entry of the matrix $V_t$. Our algorithm achieves this by quantizing the feature vectors $X_{t,a_t}$, and also the diagonal (only the diagonal) entries of the least squares matrix $V_t$. We prove that by doing so, with only $\approx 5d$ bits we can provide an unbiased estimate and guarantee an $O(\frac{1}{\sqrt{d}})$ quantization error for each entry in the matrix almost surely.

**Quantization Scheme.** We here describe the proposed quantization scheme.
• *To quantize $X_{t,a_t}$:* Let $m \triangleq \lceil\sqrt{d}\rceil$. We first send the sign of each coordinate of $X_{t,a_t}$ using $d$ bits, namely, we send the vector $s_t = X_{t,a_t}/|X_{t,a_t}|$. To quantize the magnitude $|X_{t,a_t}|$, we scale each coordinate of $|X_{t,a_t}|$ by $m$ and quantize it using a Stochastic Quantizer (SQ)[3] with $m + 1$ levels in

---

[3]As described in (3) in Section 2, SQ maps value $x$ to an integer value, namely $\lfloor x \rfloor$ with probability $\lceil x \rceil - x$ and $\lceil x \rceil$ with probability $x - \lfloor x \rfloor$.

the interval $[0, m]$. Let $X_t \triangleq \mathrm{SQ}_m(m|X_{t,a_t}|)$ denote the resulting SQ outputs, we note that $X_t$ takes non-negative integer values and lies in a norm-1 ball of radius $2d$ (this holds since the original vector lies in a norm-2 ball of radius 1 and the error in each coordinate is at most $1/m$). That is, it holds that $X_t \in \mathcal{Q} = \{x \in \mathbb{N}^d | \|x\|_1 \leq 2d\}$. We then use any enumeration $h : \mathcal{Q} \rightarrow [|\mathcal{Q}|]$ of this set to encode $X_t$ using $\log(|\mathcal{Q}|)$ bits.

• *To quantize* $X_{t,a_t} X_{t,a_t}^T$: Let $X_{t,a_t}^2$ denote a vector that collects the diagonal entries of $X_{t,a_t} X_{t,a_t}^T$. Let $\hat{X}_t \triangleq s_t X_t/m$ be the estimate of $X_{t,a_t}$ that the central learner retrieves. Note that $\hat{X}_t^2$ is not an unbiased estimate of $X_{t,a_t}^2$; however, $(X_{t,a_t}^2 - \hat{X}_t^2)_i \leq 3/m$ for all coordinates $i$ (proved in App. B). Our scheme simply conveys the difference $X_{t,a_t}^2 - \hat{X}_t^2$ with 1 bit per coordinate using a $\mathrm{SQ}_1^{[-3/m, 3/m]}$ quantizer.

The central learner and distributed agent operations are presented in Algorithm 2.

**Example 2.** Consider the case where $d = 5$. Then each coordinate of $|X_{t,a_t}|$ is scaled by 3 and quantized using $\mathrm{SQ}_3$ to one of the values $0, 1, 2, 3$ to get $X_t$. The function $h$ then maps the values for $X_t$ that satisfy the $\|X_t\|_1 \leq 10$ to a unique value (a code) in the set $[|\mathcal{Q}|]$. For instance the value $3.\mathbf{1}$ is not given a code, where $\mathbf{1}$ is the vector of all ones. However, note that for $|X_{t,a_t}|$ to be mapped to $3.\mathbf{1}$, we must have $3|(X_{t,a_t})_i| \geq 2$ for all coordinates $i$, which cannot happen since it implies that $\|X_{t,a_t}\|_2 \geq 2\sqrt{5/6} > 1$ which contradicts Assumption 1.

---

**Algorithm 2** Communication efficient for contextual linear bandits with unknown distribution

1: Input: underlying set of actions $\mathcal{A}$, and time horizon $T$.
2: $\hat{\theta}_0 = 0, \tilde{V}_0 = 0, u_0 = 0, m = \lceil \sqrt{d} \rceil$.
3: Let $h$ be an enumeration of the set $\mathcal{Q} = \{x \in \mathbb{N}^d | \|x\|_1 \leq 2d\}$.
4: **for** $t = 1 : T$ **do**
5:     **Agent**:
6:         Receive $\hat{\theta}_{t-1}$ from the central learner.
7:         Observe context realization $\{X_{t,a}\}_{a \in \mathcal{A}}$.
8:         Pull arm $a_t = \arg\max_{a \in \mathcal{A}} \langle X_{t,a}, \hat{\theta}_{t-1} \rangle$ and receive reward $r_t$.
9:         Compute the signs $s_t = X_{t,a_t}/|X_{t,a_t}|$ of $X_{t,a_t}$.
10:        Let $X_t = \mathrm{SQ}_m(m|X_{t,a_t}|)$.
11:        $e_t^2 = \mathrm{SQ}_1^{[-3/m, 3/m]}(X_{t,a_t}^2 - \hat{X}_t^2)$, where $\hat{X}_t = s_t X_t/m$.
12:        Send to the central learner $h(X_t)$, $s_t$, and $e_t^2$ using $\log_2(|\mathcal{Q}|)$, $d$, and $d$ bits, respectively.
13:        Send $\hat{r}_t = \mathrm{SQ}_1(r_t)$ using 1-bit.
14:     **Central learner**:
15:         Receive $X_t$, $s_t$, $e_t^2$, and $\hat{r}_t$ from the distributed agent.
16:         $\hat{X}_t = s_t X_t/m$, $\hat{X}_t^{(D)} = \hat{X}_t^2 + e_t^2$.
17:         $u_t \leftarrow u_{t-1} + \hat{r}_t \hat{X}_t$.
18:         $\tilde{V}_t \leftarrow \tilde{V}_{t-1} + \hat{X}_t \hat{X}_t^T - \mathrm{diag}(\hat{X}_t \hat{X}_t^T) + \mathrm{diag}(\hat{X}_t^{(D)})$.
19:         $\hat{\theta}_t \leftarrow \tilde{V}_t^{-1} u_t$.
20:         Send $\hat{\theta}_t$ to the next agent.

---

**Algorithm Performance.** Theorem 2, stated next, holds under Assumption 1 in Section 2 and some additional regulatory assumptions on the distributions $\mathcal{P}_a$ provided in Assumption 2.

**Assumption 2.** *There exist constants $c, c'$ such that for any sequence $\theta_1, ..., \theta_T$, where $\theta_t$ depends only on $H_t$, with probability at least $1 - \frac{c'}{T}$, it holds that*

$$\sum_{i=1}^t X_{i,a_i} X_{i,a_i}^T \geq \frac{ct}{d} I \quad \forall t \in [T], \tag{13}$$

*where $a_t = \arg\max_{a \in \mathcal{A}} \langle X_{t,a}, \theta_t \rangle$, and $I$ is the identity matrix.*

We note that several common assumptions in the literature imply (13), for example, bounded eigenvalues for the covariance matrix of $X_{t,a_t}$ [11, 27, 17]. Such assumptions hold for a wide range of distributions, including subgaussian distributions with bounded density [36].

**Challenge in relaxing assumption 2 (diversity assumption).** The main challenge in relaxing the diversity assumption for LinUCB (or Thompson sampling) based algorithms is that the regret of those

algorithms is bounded as $\tilde{O}(\sqrt{T}\|\hat{\theta}_T - \theta_*\|_{V_T})$. Without quantization, the quantity $\|\hat{\theta}_T - \theta_*\|_{V_T}$ grows slowly and is nearly a constant; however, without the diversity assumption, the quantization error can make $\|\hat{\theta}_T - \theta_*\|_{V_T}^2$ to grow as $\sqrt{T}$ in the worst case. This is due to the fact that sub-optimal arms do not have large number of pulls, hence, we do not have good estimate of $\theta_\star$ on those direction; on the other hand, the quantization errors in estimating $V_T$ is accumulated in all directions. As a result, the regret bound increases by a factor of $T^{1/4}$. We leave it as a future work to either relax the diversity assumption (which is required in our paper only in the case of unknown context distribution) or else show that removing it will unavoidably increase the regret order.

**Theorem 2.** *Algorithm 2 satisfies that for all $t$: $X_t \in \mathcal{Q}$; and $B_t \leq 1 + \log_2(2d+1) + 5.03d$ bits. Under assumptions 1, 2, it achieves a regret $R_T = O(d\sqrt{T \log T})$ with probability at least $1 - \frac{1}{T}$.*

**Proof Outline.** To bound the number of bits $B_t$, we first bound the size of $\mathcal{Q}$ by formulating a standard counting problem: we find the number of non-negative integer solutions for a linear equation. To bound the regret $R_T$, we start by proving that our quantization scheme guarantees some desirable properties, namely, unbiasedness and $O(\frac{1}{\sqrt{d}})$ quantization error for each vector coordinate. We then upper bound the regret in terms of $\|\hat{\theta}_t - \theta_\star\|_2$ and show that this difference can be decomposed as

$$\|\hat{\theta}_t - \theta_\star\|_2 = \|V_t^{-1}\|_2(\|\textstyle\sum_{i=1}^t E_i\|_2 + (1 + |\eta_i|)\|\textstyle\sum_{i=1}^t e_i\|_2 + \|\textstyle\sum_{i=1}^t \hat{\eta}_i X_{i,a_i}\|_2, \qquad (14)$$

where $E_t$ captures the error in estimating the matrix $X_{t,a_t} X_{t,a_t}^T$, $e_t$ is the error in estimating $X_{t,a_t}$, and $\eta_t'$ is a noise that satisfies the same properties as $\eta_t$. Using Assumption 2, we prove that $V_t^{-1}$ grows as $O(\frac{d}{t})$ with high probability, and from the unbiasdness and boundedness of all error quantities we show that they grow as $O(\sqrt{t \log t})$ with high probability. This implies that $\|\hat{\theta}_t - \theta_\star\|_2 = O(d\sqrt{\frac{\log t}{t}})$, and hence, $R_T = O(d\sqrt{T \log T})$. The complete proof is provided in App. B. $\square$

**Algorithm Complexity.** If we do not count the quantization operations, it is easy to see that the complexity of the rest of the algorithm is dominated by the complexity of computing $V_t^{-1}$ which can be done in $O(d^{2.373})$ [4]. For the quantization, we note that each coordinate of $X_t$ can be computed in $\tilde{O}(1)$ time[4]. Moreover, the computation of $h(x)$ for $x \in \mathcal{Q}$ can be done in constant time with high probability using hash tables, where $h$ is the enumeration function in Step 3. Hence, the added computational complexity is almost linear in $d$. Although a hash table for $h$ can consume $\Omega(2^{5d})$ memory, by sacrificing a constant factor in the number of bits, we can find enumeration functions that can be stored efficiently. As an example, consider the scheme in [14] that can find an one-to-one function $h : \mathcal{Q} \to \mathbb{N}^+$ which can be stored and computed efficiently, but only gives guarantees in expectation that $\mathbb{E}[\log(h(x))] = O(d)$ for all $x \in \mathcal{Q}$.

**Downlink Communication Cost.** Although we assume no-cost downlink communication, as was also the case for Algorithm 1, the downlink in Algorithm 2 is only used to send the updated parameter vector $\hat{\theta}_t$ to the agents. If desired, these estimates can be quantized using the same method as for $X_{t,a_t}$, which (following a similar proof to that of Theorem 2) can be shown to not affect the order of the regret while reducing the downlink communication to $\approx 5d$ bits per iteration.

**Offloading To Agents.** For applications where the agents wish to computationally help the central learner, the central learner may simply aggregate information to keep track of $u_t, \tilde{V}_t$ and broadcast these values to the agents; the estimate $\hat{\theta}_t$ can be calculated at each agent. Moving the computational load to the agents does not affect the regret order or the number of bits communicated on the uplink.

**Remark 2.** Under the regulatory assumptions in [17], the regret bound can be improved by a factor of $\sqrt{\log(K)/d}$, where $K = |\mathcal{A}|$ is the number of actions. However, this does not improve the regret in the worst case as the worst case number of actions is $O(C^d), C > 1$ [24].

**Societal Impact.** Results in this work can be used in decision making systems which can potentially lead to biased decisions against racial, sex, or minority groups if used without care.

**Acknowledgment.** CF and OH are supported in part by NSF award 2007714, NSF award 2221871 and Army Research Laboratory grant under Cooperative Agreement W911NF-17-2-0196. LY is supported in part by DARPA grant HR00112190130, NSF Award 2221871.

---

[4]Multiplication by $\sqrt{d}$ can take $O(\log d)$ time.

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
