# Supplementary Material of the Paper: Learning in Distributed Contextual Linear Bandits Without Sharing the Context

## A Proofs and Remarks for Section 3: Contextual Linear Bandits with Known Context Distribution

**Remark.** We note that, to reduce the multi-context problem to a single context problem in the case of known context distribution, the straightforward approach that replaces the actual context realizations $\{X_{t,a}\}_{a \in \mathcal{A}}$ with the fixed set $\{\mathbb{E}_{\mathcal{P}_a}[X_{t,a}]\}_{a \in \mathcal{A}}$, and uses this latter set as $\mathcal{X}$ in $\Lambda$, does not work and can lead to linear regret in some cases. For instance consider the case where $d = 1$, $\mathcal{A} = \{1, 2\}$, $X_{t,a} \in \{-1, 1\} \forall a \in \mathcal{A}$, $\theta_\star = 1$ and $X_{t,1}$ takes the value 1 with probability $3/4$ and $-1$ otherwise, while $X_{t,2}$ takes the values $1, -1$ with probability $1/2$. Then clearly $\langle \mathbb{E}_{\mathcal{P}_1}[X_{t,1}], \theta_\star \rangle > \langle \mathbb{E}_{\mathcal{P}_2}[X_{t,2}], \theta_\star \rangle$, however, choosing $a_t = 1 \forall t \in [T]$ leads to $\mathbb{E}[R_T] \geq T/8$ since it holds that $X_{t,1} = -1, X_{t,2} = 1$ with probability $1/8$.

**Downlink Communication.** Note that in our setup we assume that the central learner does not have any communication constraints when communicating with the distributed agents. Yet our algorithm makes frugal use of this ability: the central learner only sends the updated parameter vector $\hat{\theta}_t$. We can quantize $\hat{\theta}_t$ without performance loss if the downlink were also communication constrained using $\approx 5d$ bits and an approach similar to the one in Algorithm 2 - yet we do not expand on this in this paper, as our focus is in minimizing uplink communication costs.

### A.1 Proof of Theorem 1

**Theorem 1.** *Algorithm 1 uses* 1 *bit per reward and* 0 *bits per context. Under Assumption 1, it achieves a regret* $R_T = R_T(\Lambda) + O(\sqrt{T \log T})$ *with probability at least* $1 - \frac{1}{T}$.

**Proof.** It is obvious that the agent only sends 1 bit to the central learner to represent $r_t$ using $SQ_1$, hence, the algorithm uses 0 bits per context and 1 bit per reward. We next bound the regret of our algorithm as following. The regret can be expressed as

$$R_T = \sum_{t=1}^{T} \max_{a \in \mathcal{A}} \langle X_{t,a}, \theta_\star \rangle - \langle X_{t,a_t}, \theta_\star \rangle$$

$$= \sum_{t=1}^{T} \langle \arg\max_{X_{t,a}} \langle X_{t,a}, \theta_\star \rangle, \theta_\star \rangle - \langle \arg\max_{X_{t,a}} \langle X_{t,a}, \hat{\theta}_t \rangle, \theta_\star \rangle$$

$$= \sum_{t=1}^{T} \left( \langle \arg\max_{X_{t,a}} \langle X_{t,a}, \theta_\star \rangle, \theta_\star \rangle - \langle \mathbb{E}[\arg\max_{X_{t,a}} \langle X_{t,a}, \theta_\star \rangle], \theta_\star \rangle \right)$$

$$- \left( \langle \arg\max_{X_{t,a}} \langle X_{t,a}, \hat{\theta}_t \rangle, \theta_\star \rangle - \langle \mathbb{E}[\arg\max_{X_{t,a}} \langle X_{t,a}, \hat{\theta}_t \rangle | \hat{\theta}_t], \theta_\star \rangle \right)$$

$$+ \left( \langle \mathbb{E}[\arg\max_{X_{t,a}} \langle X_{t,a}, \theta_\star \rangle], \theta_\star \rangle - \langle \mathbb{E}[\arg\max_{X_{t,a}} \langle X_{t,a}, \hat{\theta}_t \rangle | \hat{\theta}_t], \theta_\star \rangle \right). \quad (15)$$

To bound $R_T$, we bound each of the three lines in the last expression. For the second term denoted by $\Sigma_t = \sum_{i=1}^{t} \left( \langle \arg\max_{X_{i,a}} \langle X_{i,a}, \hat{\theta}_i \rangle, \theta_\star \rangle - \langle \mathbb{E}[\arg\max_{X_{i,a}} \langle X_{i,a}, \hat{\theta}_i \rangle | \hat{\theta}_i], \theta_\star \rangle \right)$, we have that

$$\mathbb{E}[\Sigma_{t+1} | \Sigma_t] = \Sigma_t + \mathbb{E}\left[ \langle \arg\max_{X_{t,a}} \langle X_{t,a}, \hat{\theta}_t \rangle, \theta_\star \rangle - \langle \mathbb{E}[\arg\max_{X_{t,a}} \langle X_{t,a}, \hat{\theta}_t \rangle | \hat{\theta}_t], \theta_\star \rangle | \Sigma_t \right]$$

$$= \Sigma_t + \mathbb{E}\left[ \mathbb{E}\left[ \langle \arg\max_{X_{t,a}} \langle X_{t,a}, \hat{\theta}_t \rangle, \theta_\star \rangle - \langle \mathbb{E}[\arg\max_{X_{t,a}} \langle X_{t,a}, \hat{\theta}_t \rangle | \hat{\theta}_t], \theta_\star \rangle | \Sigma_t, \hat{\theta}_t \right] | \Sigma_t \right]$$

$$= \Sigma_t + \mathbb{E}\left[ \mathbb{E}\left[ \langle \arg\max_{X_{t,a}} \langle X_{t,a}, \hat{\theta}_t \rangle, \theta_\star \rangle - \langle \mathbb{E}[\arg\max_{X_{t,a}} \langle X_{t,a}, \hat{\theta}_t \rangle | \hat{\theta}_t], \theta_\star \rangle | \hat{\theta}_t \right] | \Sigma_t \right]$$

$$= \Sigma_t + \mathbb{E}\left[ \langle \mathbb{E}[\arg\max_{X_{t,a}} \langle X_{t,a}, \hat{\theta}_t \rangle | \hat{\theta}_t], \theta_\star \rangle - \langle \mathbb{E}[\arg\max_{X_{t,a}} \langle X_{t,a}, \hat{\theta}_t \rangle | \hat{\theta}_t], \theta_\star \rangle | \Sigma_t \right]$$

$$= \Sigma_t. \tag{16}$$

We also have that

$$\left|\Sigma_t - \Sigma_{t-1}\right| \leq \left\|\langle \arg\max_{X_{t,a}}\langle X_{t,a}, \hat{\theta}_t\rangle, \theta_\star\rangle - \langle \mathbb{E}[\arg\max_{X_{t,a}}\langle X_{t,a}, \hat{\theta}_t\rangle], \theta_\star\rangle\right\| \|\theta_\star\|$$

$$\leq \left\|\langle \arg\max_{X_{t,a}}\langle X_{t,a}, \hat{\theta}_t\rangle, \theta_\star\rangle\right\| + \left\|\langle \mathbb{E}[\arg\max_{X_{t,a}}\langle X_{t,a}, \hat{\theta}_t\rangle], \theta_\star\rangle\right\|$$

$$\leq \left\|\arg\max_{X_{t,a}}\langle X_{t,a}, \hat{\theta}_t\rangle\right\|\|\theta_\star\| + \left\|\mathbb{E}[\arg\max_{X_{t,a}}\langle X_{t,a}, \hat{\theta}_t\rangle]\right\|\|\theta_\star\| \leq 2. \tag{17}$$

Hence, $\Sigma_t$ is a martingale with bounded difference. By Azuma–Hoeffding inequality [45], we have that $\|\Sigma_T\| \leq C\sqrt{T\log T}$ with probability at least $1 - \frac{1}{2T}$. Similarly, the first line in (15) is a martingale with bounded difference, hence, the following holds with probability at least $1 - \frac{1}{2T}$

$$\left|\sum_{t=1}^{T}\langle \arg\max_{X_{t,a}}\langle X_{t,a}, \theta_\star\rangle, \theta_\star\rangle - \langle \mathbb{E}[\arg\max_{X_{t,a}}\langle X_{t,a}, \theta_\star\rangle], \theta_\star\rangle\right| \leq C\sqrt{T\log T}. \tag{18}$$

By substituting in (15) and using the union bound we get that the following holds with probability at least $1 - \frac{1}{T}$

$$R_T \leq C\sqrt{T\log T} + \sum_{t=1}^{T}\left(\langle \mathbb{E}[\arg\max_{X_{t,a}}\langle X_{t,a}, \theta_\star\rangle], \theta_\star\rangle - \langle \mathbb{E}[\arg\max_{X_{t,a}}\langle X_{t,a}, \hat{\theta}_t\rangle|\hat{\theta}_t], \theta_\star\rangle\right)$$

$$= C\sqrt{T\log T} + \sum_{t=1}^{T}\langle X^*(\theta_\star), \theta_\star\rangle - \langle X^*(\hat{\theta}_t), \theta_\star\rangle$$

$$= C\sqrt{T\log T} + \sum_{t=1}^{T}\langle X^*(\theta_\star), \theta_\star\rangle - \langle X_t, \theta_\star\rangle. \tag{19}$$

We also have by definition of $X^*(\theta_\star)$ that for any given $\theta, \theta_\star$

$$\langle X^*(\theta_\star), \theta_\star\rangle = \mathbb{E}[\max_{X_{t,a}}\langle X_{t,a}, \theta_\star\rangle]$$

$$\geq \mathbb{E}[\langle \arg\max_{X_{t,a}}\langle X_{t,a}, \theta\rangle, \theta_\star\rangle] = \langle X^*(\theta), \theta_\star\rangle. \tag{20}$$

Hence, we have that $\max_{X\in\mathcal{X}}\langle X, \theta_\star\rangle = \langle X^*(\theta_\star), \theta_\star\rangle$. By substituting in (19), we get that

$$R_T \leq C\sqrt{T\log T} + \sum_{t=1}^{T}\max_{X\in\mathcal{X}}\langle X, \theta_\star\rangle - \langle X_t, \theta_\star\rangle = C\sqrt{T\log T} + R_T(\Lambda), \tag{21}$$

where $R_T(\Lambda)$ is the regret of the subroutine $\Lambda$. $\qquad\square$

### A.2  Proof of Corollary 1

**Corollary 1.** Suppose we are given $\tilde{X}^\star$ that satisfies (12). Then, there exists an algorithm $\Lambda$ for which Algorithm 1 achieves $R_T = \tilde{O}(d\sqrt{T} + \epsilon T\sqrt{d})$ with probability at least $1 - \frac{1}{T}$.

**Proof.** $\Lambda$ assumes that the reward $r_t$ is generated according to $\langle \tilde{X}^\star(\hat{\theta}_t), \theta_\star\rangle + \eta_t$, while it is actually generated according to

$$r_t = \langle X^\star(\hat{\theta}_t), \theta_\star\rangle + \eta_t = \langle \tilde{X}^\star(\hat{\theta}_t), \theta_\star\rangle + \eta_t + f(\hat{\theta}_t), \tag{22}$$

where $f(\hat{\theta}_t) = \langle X^\star(\hat{\theta}_t) - \tilde{X}^\star(\hat{\theta}_t), \theta_\star\rangle$. We have that

$$|f(\hat{\theta}_t)| \leq \|X^\star(\hat{\theta}_t) - \tilde{X}^\star(\hat{\theta}_t)\|\|\theta_\star\| \leq \epsilon. \tag{23}$$

Hence, the rewards follow a misspecified linear bandit model [24]. It was shown in [24] that for the single context case, there is an algorithm $\Lambda$ that achieves $R_T(\Lambda) = \tilde{O}(d\sqrt{T} + \epsilon T)$ with probability at least $1 - \frac{1}{T}$. The corollary follows from Theorem 1 by noting that $R_T(\Lambda)$ is defined based on the true $X^\star$ as in (5). $\qquad\square$

# B  Proofs of Section 4: Contextual Linear Bandits with Unknown Context Distribution

## B.1  Proof of Theorem 2

**Theorem 2.** *Algorithm 2 satisfies that for all $t$: $X_t \in \mathcal{Q}$; and $B_t \leq 1 + \log_2(2d+1) + 5.03d$ bits. Under assumptions 1, 2, it achieves a regret $R_T = O(d\sqrt{T \log T})$ with probability at least $1 - \frac{1}{T}$.*

**Proof.** We start by proving some properties about the quantized values $\hat{X}_t, \hat{r}_t, \hat{X}_t^2$. We first note that be definition of SQ, we have that

$$m|(\hat{X}_t - X_{t,a_t})_j| \leq 1. \tag{24}$$

Hence,

$$|(\hat{X}_t^2 - X_{t,a_t}^2)_j| = |(\hat{X}_t^2 - (\hat{X}_t + X_{t,a_t} - \hat{X}_t)^2)_j| = |(2(X_{t,a_t} - \hat{X}_t)\hat{X}_t + (X_{t,a_t} - \hat{X}_t)^2)_i|$$

$$\leq 2|(X_{t,a_t} - \hat{X}_t)_i||(\hat{X}_t)_i| + |((X_{t,a_t} - \hat{X}_t)^2)_i| \leq \frac{2}{m} + \frac{1}{m^2} \leq \frac{3}{m}. \tag{25}$$

We also have that

$$\mathbb{E}[\hat{X}_t^{(D)}|X_{t,a_t}^2] = \mathbb{E}[\hat{X}_t^2 + e_t^2|X_{t,a_t}^2] = \mathbb{E}[\mathbb{E}[\hat{X}_t^2 + e_t^2|X_{t,a_t}^2, \hat{X}_t]|X_{t,a_t}^2]$$

$$= \mathbb{E}[\hat{X}_t^2 + \mathbb{E}[e_t^2|X_{t,a_t}^2 - \hat{X}_t^2]|X_{t,a_t}^2] = \mathbb{E}[\hat{X}_t^2 + X_{t,a_t}^2 - \hat{X}_t^2|X_{t,a_t}^2] = X_{t,a_t}^2. \tag{26}$$

In summary, from this and the definition of SQ, we get that

$$m|(\hat{X}_t - X_{t,a_t})_j| \leq 1, \mathbb{E}[\hat{X}_t|X_{t,a_t}] = X_{t,a_t}$$

$$m|\hat{X}_t^{(D)} - X_{t,a_t}^2| \leq 3, \mathbb{E}[\hat{X}_t^{(D)}|X_{t,a_t}^2] = X_{t,a_t}^2$$

$$|\hat{r}_t - r_t| \leq 1, \mathbb{E}[\hat{r}_t|r_t] = r_t \tag{27}$$

We next show that $X_t \in \text{dom}(h)$. By definition of SQ, we have that $X_t \in \mathbb{N}^d$. We also have that

$$\|X_t\|_1 = \sum_{i=1}^{d} m|(X_{t,a_t})_i| \leq 1 + \lfloor \sqrt{d}|(X_{t,a_t})_i| \rfloor \leq d + \sum_{i=1}^{d} \lfloor \sqrt{d}|(X_{t,a})_i| \rfloor^2$$

$$\leq d + d\|X_{t,a_t}\|^2 \leq 2d. \tag{28}$$

Therefore, we have that $X_t \in \mathcal{Q} = \text{dom}(h)$.

We next show the upper bound on the number of bits $B_t$. We have that $\hat{r}_t$ uses 1 bit, the sign vector $s_t$ uses $d$ bits, $e_t^2$ uses $d$ bits and $X_t$ uses $\log(|\mathcal{Q}|)$ bits. We bound $|\mathcal{Q}|$ as follows. The number of non-negative solutions for the equation $\|a\|_1 = x$ for $a \in \mathbb{N}^d, x \in \mathbb{N}$ is $\binom{d+x-1}{x} \leq \binom{d+x}{x} = \binom{d+x}{d}$, hence,

$$|\mathcal{Q}| \leq (2d+1)\binom{3d}{d} \leq (2d+1)(e\frac{3d}{d})^d. \tag{29}$$

Hence, we have that

$$B_t \leq 1 + \log(2d+1) + (2 + \log(3e))d. \tag{30}$$

We next show the regret bound. We start by bounding the regret in iteration $t$ by the distance between $\theta_\star, \hat{\theta}_{t-1}$. From step 7 of Algorithm 2, we have that $\langle X_{t,a_t}, \hat{\theta}_{t-1} \rangle \geq \langle X_{t,a}, \hat{\theta}_{t-1} \rangle \forall a \in \mathcal{A}$, hence, we have that

$$\max_{a \in \mathcal{A}} \langle X_{t,a}, \theta_\star \rangle - \langle X_{t,a_t}, \theta_\star \rangle \leq \max_{a \in \mathcal{A}} \langle X_{t,a} - X_{t,a_t}, \theta_\star \rangle - \max_{a \in \mathcal{A}} \langle X_{t,a} - X_{t,a_t}, \hat{\theta}_{t-1} \rangle$$

$$\leq \max_{a \in \mathcal{A}} \|X_{t,a} - X_{t,a_t}\|\|\theta_\star - \hat{\theta}_{t-1}\| \leq 2\|\theta_\star - \hat{\theta}_{t-1}\|. \tag{31}$$

We next bound the distance $\|\theta_\star - \hat{\theta}_{t-1}\|$. Let us denote $e_t = \hat{X}_t - X_{t,a_t}, \hat{\eta}_t = \eta_t + (\hat{r}_t - r_t), E_t = X_{t,a_t}X_{t,a_t}^T - (V_t - V_{t-1})$. We have that

$$\|\theta_\star - \hat{\theta}_t\| = \|\theta_\star - V_t^{-1}\sum_{i=1}^{t} \hat{r}_i\hat{X}_i\| = \|\theta_\star - V_t^{-1}\sum_{i=1}^{t}(X_{i,a_i}X_{i,a_i}^T\theta_\star + r_ie_i + \hat{\eta}_iX_{i,a_i} + \hat{\eta}_ie_i)\|$$

$$= \|V_t^{-1} \sum_{i=1}^{t} (E_i \theta_\star + r_i e_i + \hat{\eta}_i X_{i,a_i} + \hat{\eta}_i e_i)\|$$

$$\leq \|V_t^{-1}\| (\|\sum_{i=1}^{t} E_i\| + (|r_i| + |\eta_i|)\|\sum_{i=1}^{t} e_i\| + \|\sum_{i=1}^{t} \hat{\eta}_i X_{i,a_i}\|$$

$$\leq \|V_t^{-1}\| (\|\sum_{i=1}^{t} E_i\| + (1 + |\eta_i|)\|\sum_{i=1}^{t} e_i\| + \|\sum_{i=1}^{t} \hat{\eta}_i X_{i,a_i}\|. \tag{32}$$

We next bound each of the values in the last expression. As $\eta_i$ is subgaussian we have that with probability at least $1 - \frac{1}{5T^2}$, we have that $|\eta_i| \leq C \log T \forall i \in [T]$. We also have that, using (27), $S_t^e = \sum_{i=1}^{t} e_i$ is a martingale with bounded difference, hence, by Azuma–Hoeffding inequality, we get that with probability at least $1 - \frac{1}{5dT^2}$ we have that $|(S_t^e)_j| \leq \frac{C}{\sqrt{d}} \sqrt{t \log(dT)}$; note that $|(e_t)_i| \leq \frac{1}{\sqrt{d}}$. Hence, by the union bound we get that with probability at least $1 - \frac{1}{5T^2}$ we have that $\|\sum_{i=1}^{t} e_i\| \leq C \sqrt{t \log(dT)}$. Similarly, conditioned on $X_{1,a_1}, ..., X_{t,a_t}, \sum_{i=1}^{t} \hat{\eta}_i X_{i,a_i}$ is a martingale with bounded difference, hence, with probability at least $1 - \frac{1}{5dT^2}$ we have that $|(\sum_{i=1}^{t} \hat{\eta}_i X_{i,a_i})_j| \leq C \sqrt{\sum_{i=1}^{t} (X_{i,a_i})_j^2 \log(dT)}$. Hence, with probability at least $1 - \frac{1}{5T^2}$ we have that $\|\sum_{i=1}^{t} \hat{\eta}_i X_{i,a_i}\| \leq C \sqrt{\sum_{i=1}^{t} \|X_{i,a_i}\|^2 \log(dT)} \leq C \sqrt{t \log(dT)}$. Summing up, we get that with probability at least $1 - \frac{3}{5T^2}$

$$\|\theta_\star - \hat{\theta}_t\| \leq \|V_t^{-1}\| (\|\sum_{i=1}^{t} E_i\| + C \sqrt{t \log(dT)}). \tag{33}$$

It remains to bound $\|V_t^{-1}\|, \|\sum_{i=1}^{t} E_i\|$ which we do in the following by starting with $\|\sum_{i=1}^{t} E_i\|$. We have that

$$E_i = X_{i,a_i} X_{i,a_i}^T - \hat{X}_i \hat{X}_i^T + \text{diag}(\hat{X}_i \hat{X}_i^T) - \text{diag}(\hat{X}_t^{(D)})$$

$$= \text{diag}(\hat{X}_i \hat{X}_i^T) - 2 X_{i,a_i} e_i^T - e_i e_i^T - \text{diag}(\hat{X}_t^{(D)})$$

$$= 2\text{diag}(X_{i,a_i} e_i^T) - 2 X_{i,a_i} e_i^T - (e_i e_i^T - \text{diag}(e_i e_i^T)) - \text{diag}(\hat{X}_t^{(D)} - X_{i,a_i}^2). \tag{34}$$

Hence, we have that

$$\|\sum_{i=1}^{t} E_i\| \leq 2\|\sum_{i=1}^{t} \text{diag}(X_{i,a_i} e_i^T)\| + 2\|\sum_{i=1}^{t} X_{i,a_i} e_i^T\|$$

$$+ \|\sum_{i=1}^{t} e_i e_i^T - \text{diag}(e_i e_i^T)\| + \|\sum_{i=1}^{t} \text{diag}(\hat{X}_t^{(D)} - X_{i,a_i}^2)\|. \tag{35}$$

We have that, using (27), conditioned on $X_{1,a_1}, ..., X_{t,a_t}, \sum_{i=1}^{t} \text{diag}(X_{i,a_i} e_i^T)$ is a martingale with bounded difference, hence, similar to what we did before using Azuma–Hoeffding inequality and the union bound we get that with probability at least $1 - \frac{1}{20T^2}$, we have that $\|\sum_{i=1}^{t} \text{diag}(X_{i,a_i} e_i^T)\| \leq \frac{C}{\sqrt{d}} \sqrt{t \log(dT)}$. Similarly, with probability at least $1 - \frac{1}{20T^2}$, we have that $\|\sum_{i=1}^{t} \text{diag}(\hat{X}_t^{(D)} - X_{i,a_i}^2)\| \leq \frac{C}{\sqrt{d}} \sqrt{t \log(dT)}$. We next turn to bounding $\|\sum_{i=1}^{t} X_{i,a_i} e_i^T\|$. Conditioned on $X_{1,a_1}, ..., X_{t,a_t}$, we have that by Azuma–Hoeffding, with probability at least $1 - \frac{1}{d^2 T^2}$, we have

$$|(\sum_{i=1}^{t} X_{i,a_i} e_i^T)_{jk}| \leq \frac{C}{\sqrt{d}} \sqrt{\sum_{i=1}^{t} (X_{i,a_i})_j^2 \log(dT)}. \tag{36}$$

We notice that taking the absolute value of all elements of a matrix does not decrease its maximum eigenvalue, hence, by the union bound we have that with probability at least $1 - \frac{1}{20T^2}$ we have that

$$\|\sum_{i=1}^{t} X_{i,a_i} e_i^T\| \leq \frac{C\sqrt{\log(dT)}}{\sqrt{d}} \|\mathbf{1}[\sqrt{\sum_{i=1}^{t} (X_{i,a_i})_1^2}, ..., \sqrt{\sum_{i=1}^{t} (X_{i,a_i})_d^2}]\|$$

$$\leq \frac{C\sqrt{\log(dT)}}{\sqrt{d}}\sqrt{d\sum_{i=1}^{t}\|X_{i,a_i}\|^2} \leq C\sqrt{t\log(dT)}. \tag{37}$$

To bound $\|\sum_{i=1}^{t}e_i e_i^T - \mathrm{diag}(e_i e_i^T)\|$, we notice that for all elements except the diagonal we have that $\mathbb{E}[(e_i)_j(e_i)_k] = \mathbb{E}[(e_i)_j(e_i)_k|X_{t,a_t}] = \mathbb{E}[(e_i)_k|X_{t,a_t}]\mathbb{E}[(e_i)_j|X_{t,a_t}] = 0, j \neq k$. Hence, it can be shown that $\sum_{i=1}^{t}(e_i)_j(e_i)_k$ is a martingale with bounded difference for $j \neq k$, hence, with probability at least $1 - \frac{1}{20d^2T^2}$, we have that $|\sum_{i=1}^{t}(e_i)_j(e_i)_k| \leq \frac{C}{d}\sqrt{t\log(dT)}$. Hence, by the union bound we get that with probability at least $1 - \frac{1}{20T^2}$

$$\|\sum_{i=1}^{t}e_i e_i^T - \mathrm{diag}(e_i e_i^T)\| \leq \frac{C\sqrt{t\log(dT)}}{d}\|\mathbf{1}\mathbf{1}^T\| \leq C\log(dT). \tag{38}$$

Hence, from (35) and the union bound we have that with probability at least $1 - \frac{1}{5T^2}$

$$\|\sum_{i=1}^{t}E_i\| \leq C\sqrt{t\log(dT)}. \tag{39}$$

We next turn to bounding $\|V_t^{-1}\|$. We have from (39), and Assumption 2, and the union bound, the following holds with probability at least $1 - \frac{2}{5T^2}$

$$\|V_t\| = \|\sum_{i=1}^{t}X_{i,a_i}X_{i,a_i}^T - E_i\| \geq \|\sum_{i=1}^{t}X_{i,a_i}X_{i,a_i}^T\| - \|E_i\| \geq C(\frac{t}{d} - \sqrt{t\log(dT)}). \tag{40}$$

Hence for $t \geq 4\log(dT)$, we have that with probability at least $1 - \frac{2}{5T^2}$, it holds that $\|V_t\| \geq C\frac{t}{2d}$, and hence,

$$\|V_t^{-1}\| \leq C\frac{d}{t}. \tag{41}$$

Hence, from (32) and the union bound, the following holds with probability at least $1 - \frac{1}{T^2}$

$$\|\theta_\star - \hat{\theta}_t\| \leq Cd\frac{\sqrt{\log(dT)}}{\sqrt{t}}. \tag{42}$$

Therefore, from (31) and the union bound again we have that the following holds with probability at least $1 - \frac{1}{T}$

$$R_T \leq \sum_{t=1}^{T}Cd\frac{\sqrt{\log(dT)}}{\sqrt{t}} \leq Cd\sqrt{\log(dT)}(1 + \int_{t=1}^{T}\frac{1}{\sqrt{t}}dt)$$
$$\leq 2Cd\sqrt{T\log(dT)}. \tag{43}$$

$\square$