# OpenReview forum: "Learning from Distributed Users in Contextual Linear Bandits Without Sharing the Context"
_NeurIPS.cc/2022/Conference — NeurIPS 2022 Accept_

### Official Review · Reviewer_iMrk · 2022-07-10

**Rating:** 5
**Confidence:** 3
**Soundness:** 2 fair
**Presentation:** 3 good
**Contribution:** 3 good

**Summary:**

This paper considers the contextual bandit with limited communication. In this problem, each arm has a context distribution and a context at each round t=1,2,...,T is iid from the corresponding distribution. The goal is to learn the coefficient theta* to choose the arm of the largest reward rti = <Xti, theta_*> + eta_t, where eta_t is an iid noise. For ease of discussion, all contexts and theta_* lie in a unit ball. It is well-known that UCB and Thompson sampling algorithms are effective in this setting. Each agent receives the estimated theta from the central learner, chooses an arm, and sends information to the learner. Regarding the learner's ability, this paper considers two settings: First setting is that the learner knows the context distribution. The second setting is that the learner does not know the context distribution.

Algorithm 1 for the first setting sends 1-bit information that suffices to have optimal sqrt{T log T} rate. Algorithm 2 for the second setting sends 5d-bit information and has the optimal rate.

This paper is mostly well-written. The introduction of algorithms can be improved. For example, Line154-160 and L194-206 are not very helpful before I actually see the algorithm with example. I feel the results somewhat lack integration (for example, alg 1 is black-box whereas alg 2 is base algorithm dependent), but overall the results are slightly above the bar.

Details:

Algorithm 1 is a meta-algorithm that internally uses the standard bandit algorithm \Lambda. The crux in this part is that, if the algorithm sends the agent hattheta_t = X^{-1}(xt), then the average context is xt and the 1-bit signal is an unbiased estimator of rt = <xt, theta>. Since agents never send contextual information to the learner, this algorithm depends on the context distribution, but some remedy is discussed in Line272-274.

Algorithm 2 is an algorithm that sends partial information about the contexts. The algorithm carefully quantifies contexts. Note that if several terms are multiplicated in the update formula, the composite terms also need to be quantified. It assumes that every path of arms linearly increases each eigenvalue and the exploration-exploitation tradeoff is not seriously dealt.

The model depends on the assumption that rt is bounded, and extending it to Gaussian bandits is plausible.


**Questions:**

* Is the computational complexity of alg 1 addressed?
* Is it possible to analyze incorrectly specified context distribution?
* Can we make algorithm 2 black-box?
* What happens when the minimum eigenvalue is not guaranteed to grow in algorithm 2?
* Is there any justification for the limited uplink communication while allowing unlimited downlink communication?

**Limitations:**

I found no ethical problem.
Algorithm 2 omits the exploration-exploitation tradeoff, but is okay for the conceptual idea for these results.

**Strengths And Weaknesses:**

Strengths:
Elegant setup and algorithms.
robustness to the estimation error of context distribution.

Weaknesses:
Inpractical settings (unlimited downlink communication).
Insufficient discussion on the operation complexity of obtaining X^*(theta) and its inverse (alg 1).

---

> ### Author Response · Authors · 2022-08-02
> **$\textbf{[Part 2]}$ Diversity assumption, unlimited downlink, relax bounded $r_t$**
>
> $\textbf{[Diversity assumption.]}$ Indeed,  relaxing the diversity assumption (which is required in our paper only in the case of unknown context distribution) is an interesting direction for future work we had also identified. Since the central learner can broadcast parameters to the agents, the central learner can also coordinate the exploration. However, the main challenge that we faced in relaxing the diversity assumption for LinUCB-based algorithms is that the regret of LinUCB is bounded as $\tilde{O}(\sqrt{T}\\|\hat{\theta_T}-\theta_*\\|_{V_T})$.
>
> Without quantization, the quantity $\\|\hat{\theta_T}-\theta_*\\|_{V_T}$ grows slowly and is nearly a constant;
>
> however, without the diversity assumption, the quantization error can make $\\|\hat{\theta_T}-\theta_*\\|^2_{V_T}$ to grow as $\sqrt{T}$ in the worst case. This results in an extra factor of $T^{1/4}$ in the regret bound. The same problem persists if we use Thompson sampling. We are not sure if this is unavoidable or there are other analysis techniques that can result in better regret bounds.
>
> As other works that started with the diversity assumption, e.g., see [1-4] below, we hope this assumption can either be relaxed (for the case where the context distribution is unknown, perhaps assuming some partial knowledge of the distribution) in a future work, or if not to show that removing it will unavoidably increase the regret order.
> We are happy to include a discussion along the above lines in the revised paper.
>
> [1] I. Bogunovic, A. Losalka, A. Krause, and J. Scarlett, “Stochastic linear bandits robust to adversarial attacks,” in International Conference on Artificial Intelligence and Statistics. PMLR, 2021, pp. 991–999.
>
> [2] W. Wu, J. Yang, and C. Shen, “Stochastic linear contextual bandits with diverse contexts,” in International Conference on Artificial Intelligence and Statistics. PMLR, 2020, pp. 2392–2401.
>
> [3] M. Papini, A. Tirinzoni, M. Restelli, A. Lazaric, and M. Pirotta, “Leveraging good representations in linear contextual bandits,” in International Conference on Machine Learning. PMLR, 2021, pp. 8371–8380.
>
> [4] L. Li, Y. Lu, and D. Zhou, “Provably optimal algorithms for generalized linear contextual bandits,” in International Conference on Machine Learning. PMLR, 2017, pp. 2071–2080.
>
>
> $\textbf{[Downlink communication.]}$ Indeed the focus in this paper is on  algorithms that optimize the uplink communication (from the agents to the learner), and assume unlimited (cost-free) downlink communication. This is a standard assumption in wireless communication (e.g. see [1, 2, 3] below) for several reasons: uplink wireless links tend to be much more bandwidth restricted, since several users may be sharing the same channel as the central learner can be doing multiple tasks, and hence, may connect with multiple users simultaneously; uplink communication may also be battery-powered and thus more expensive to sustain; moreover, the agents may be less motivated to expend communication energy than the central learner who benefits from learning. Having said that,
> we note that we can quantize $\hat{\theta}$ without performance loss if the downlink were also communication constrained using $\approx 5d$ bits similar to quantizing $X_{t,a_t}$ in Algorithm 2. The analysis in the case of unknown context distribution is similar to the proof of Theorem 2.
>
> [1] M. H. Anisi, G. Abdul-Salaam, and A. H. Abdullah. A survey of wireless sensor network approaches and their energy consumption for monitoring farm fields in precision agriculture. Precision Agriculture, 16(2):216–238, 2015.
>
> [2] T. D. Novlan, H. S. Dhillon, and J. G. Andrews. Analytical modeling of uplink cellular networks. IEEE Transactions on Wireless Communications, 12(6):2669–2679, 2013.
>
> [3] O. A. Hanna, L. Yang, and C. Fragouli. Solving multi-arm bandit using a few bits of communication. In International Conference on Artificial Intelligence and Statistics, pages 11215–11236. PMLR, 2022.
>
>
> $\textbf{[Extension to relax bounded $r_t$.]}$ Yes indeed, the boundedness assumption of $r_t$ can be relaxed using a reward compression framework in [16], which only requires approximately $3.5$ bits on average to send ths $r_t$'s, even if it is unbounded. We are happy to include this observation to our paper.

---

> ### Author Response · Authors · 2022-08-02
> **$\textbf{[Part 1]}$ Introduction of algorithms, complexity of $X^*(\theta)$, incorrectly specified context distribution, Algorithm 2 to rely on black-box**
>
> We thank the reviewer for the thorough review and for appreciating our contribution.
>
> $\textbf{[Introduction of Algorithms.]}$ We are happy to rewrite the introduction of the Algorithms to more clearly explain their main ideas.
>
>
> $\textbf{[Complexity of computing $X^*(\theta)$.]}$ Good point! The time-complexity of $X^*(\theta)$ depends on the context distribution. Indeed, we agree that calculating $X^*(\theta)$ can be computationally expensive in worst-case scenarios. However, we believe that for many practical distribution, $X^*(\theta)$ can be computed/approximated efficiently; we here give more examples:
>
> (1) If $\\{P_a\\}_{a\in \mathcal{A}}$ are continuous distributions, then $X^*(\theta), \theta \neq 0$
> can be expressed as
>
> $X^*(\theta)=\sum_{a\in \mathcal{A}} \int_{x_a\sim P_a}x_a E_{x_{a'}\sim P_{a'},a'\in \mathcal{A}/ \\{a\\}}[I[\langle x_{a'},\theta\rangle < \langle x_{a},\theta\rangle \forall a'\neq a]|x_a]dP_a$.
>
> For many distributions, the previous quantity can be computed/approximated efficiently, e.g., consider the case where $d\geq 1,\ x_a$ are iid $d$-dimensional Gaussian vectors with mean $\mu$ and covariance matrix $\Sigma = U^TDU$, where $D$ is a diagonal matrix and $U$ is upper triangular. The expectation in the previous formula is equal to $(Q(\frac{\langle x_a-\mu, \theta \rangle}{\\|\sqrt{D}U\theta\\|_2}))^{|\mathcal{A}|-1}$,
>
> where $Q(c)=\frac{1}{\sqrt{2\pi}}\int_{c}^\infty \exp(-\frac{1}{2}x^2)dx$, and hence, $X^*(\theta)$ can be approximated efficiently in that case.
>
> (2) For discrete distributions, $X^*(\theta)$ can be computed efficiently depending on the number of mass points of the distribution and if the distribution has structures/properties to simplify the expression.
>
>
> We hope that our result, which shows the possibility of learning without sharing the context at all, may motivate and inspire other works to provide efficient algorithms for a larger class of problems.
>
> We are happy to include this discussion  in a revised version of our paper.
>
>
> $\textbf{[Incorrectly specified context distribution.]}$ We provide a first such result in Corollary 1. We leave it as a future direction to prove whether the result in Corollary 1 is tight, or can be improved.
>
>
>
> $\textbf{[Algorithm 2 to rely on black-box.]}$ We believe it is non-trivial to make Algorithm 2 rely on a black-box algorithm, and could be a separate publication. The challenge we faced is that the error in estimating the feature vectors $X_{t,a_t}$ will affect the performance differently depending on how the black-box algorithm processes feature vectors. One approach we thought about is to come up with a quantization scheme that satisfies the following properties: $\|Q(X_{t,a_t})-X_{t,a_t}\|\leq  C, \mathbb{E}[X_{t,a_t}|Q(X_{t,a_t})]=Q(X_{t,a_t})$. This can be shown to enable Algorithm 2 to rely on a black-box algorithm, however, we believe those properties cannot be satisfied for a quantizer that uses a finite number of bits and $X_{t,a_t}\in \mathbb{R}^d$. This is because the prior distribution of $X_{t,a_t}$ (which we cannot control) will affect $\mathbb{E}[X_{t,a_t}|Q(X_{t,a_t})]$. Thus this remains an open question.

---

### Official Review · Reviewer_y1xV · 2022-07-10

**Rating:** 6
**Confidence:** 3
**Soundness:** 3 good
**Presentation:** 3 good
**Contribution:** 2 fair

**Summary:**

This paper studies a distributed linear bandit setting with a central learner and agents that observe contexts and execute decisions. As the central learner does not observe the context, each agent needs to communicate its decision and observations. The goal of this paper is to minimize "uplink" communication, i.e. the number of bits each agents needs to transfer in order to enable central low regret learning. Two results are provided under different assumptions on the context distribution. The proposed solution is a reduction framework that leverages existing results on linear bandits.

**Questions:**

* How is (6) defined for parameters with multiple optimal actions? For instance at $\\theta = 0$, all actions are optimal, and the definition of $X^*(\\theta)$ depends on this choice.



**Limitations:**

* It is unclear if $X^*(\\theta)$ and its inverse can be computed. The provided example is for d=1, whereas d>1 will be relevant for many applications. This is discussed around 267-271, but I do not see that the authors provide a good resolution. This is potentially quite a big limitation of the method.

* The regret bound for unknown contextual distribution requires a strong diversity assumption (Assumption 2). This allows the agents to essentially always play the greedy action. Without this assumption, the agents would need to engage in active exploration, and it is perhaps less clear how to coordinate this. A straightforward idea could be to use Thompson sampling, and communicate the sampled parameter to the agent.

**Strengths And Weaknesses:**

**Clarity:**

Overall, the paper is well written. I have some minor remarks to further improve clarity:

* line 66 and 300: Please discuss why previous works require O(d log(T)) bits and explain what is meant by "exponential complexity". Note that the reference [19] does not provide algorithms for the distributed setting, as far as I know.
* line 91: Are two different notations needed for vector indices? Consider using only one.
* line 120: The cited bounds are not quite correct; [1] achieves $d \\sqrt(T) \\log(T)$; [28] achieves $d \\sqrt(T) \\log(T)^{3/2}$; [2] achieves roughly $d \\sqrt{T \\log(K)}$ (which is a factor $\\log(K)$ worse). Moreover, these bounds are not the best known, see e.g. https://arxiv.org/abs/1904.00242 and https://arxiv.org/abs/1905.01435
* below line 190: I find the notation $\\theta^*(\\Lambda)$ very confusing: Why is the unknown parameter a function of the algorithm?
* line 216: It might hep to say that the agent plays the argmax action for $\\hat \\theta_t$!
* line 220: In what sense do we have $\\theta_t = \\theta^*$? As written this is almost certainly not true.
* Algorithm 1 / 2: Consider using the same ordering for central learner and agents in the pseudo code
* Eq (14) in the appendix: Why do you introduce a conditioning on $\\theta^*$ in the expectation? $\theta^*$ is not a random variable.

**Quality:**

The assumptions and results are clearly stated. I liked that the main paper provides proof sketches. I spent some time checking the proofs in the appendix and did not identify any technical flaws.

**Originality & Significance:**

This paper studies a well motivated setting. Related work is discussed. I am not an expert in distributed computation, but to me the ideas in the context of linear bandits are novel.

**Minor:**

* 191: typo in 'uknown'
* 254: typo in 'unbiasdness'

---

> ### Author Response · Authors · 2022-08-02
> **$\textbf{[Part 2]}$ Ties in $X^*(\theta)$, complexity of $X^*(\theta)$, diversity assumption**
>
> $\textbf{[Ties in $X^*(\theta)$]}$ Ties can be broken uniformly at random. Any pre-selected choice function (e.g., by ordering the vectors lexicographically and choosing the vector with minimum order) would work as long as the same function is used in step 12 of Algorithm 1.
>
> $\textbf{[Complexity of computing $X^*(\theta)$.]}$ Good point! The time-complexity of $X^*(\theta)$ depends on the context distribution. Indeed, we agree that calculating $X^*(\theta)$ can be computationally expensive in worst-case scenarios. However, we believe that for many practical distribution, $X^*(\theta)$ can be computed/approximated efficiently; we here give more examples:
>
> (1) If $\\{P_a\\}_{a\in \mathcal{A}}$ are continuous distributions, then $X^*(\theta), \theta \neq 0$
> can be expressed as
>
> $X^*(\theta)=\sum_{a\in \mathcal{A}} \int_{x_a\sim P_a}x_a E_{x_{a'}\sim P_{a'},a'\in \mathcal{A}/ \\{a\\}}[I[\langle x_{a'},\theta\rangle < \langle x_{a},\theta\rangle \forall a'\neq a]|x_a]dP_a$.
>
> For many distributions, the previous quantity can be computed/approximated efficiently, e.g., consider the case where $d\geq 1,\ x_a$ are iid $d$-dimensional Gaussian vectors with mean $\mu$ and covariance matrix $\Sigma = U^TDU$, where $D$ is a diagonal matrix and $U$ is upper triangular. The expectation in the previous formula is equal to $(Q(\frac{\langle x_a-\mu, \theta \rangle}{\\|\sqrt{D}U\theta\\|_2}))^{|\mathcal{A}|-1}$,
>
> where $Q(c)=\frac{1}{\sqrt{2\pi}}\int_{c}^\infty \exp(-\frac{1}{2}x^2)dx$, and hence, $X^*(\theta)$ can be approximated efficiently in that case.
>
> (2) For discrete distributions, $X^*(\theta)$ can be computed efficiently depending on the number of mass points of the distribution and if the distribution has structures/properties to simplify the expression.
>
>
> We hope that our result, which shows the possibility of learning without sharing the context at all, may motivate and inspire other works to provide efficient algorithms for a larger class of problems.
>
> We are happy to include this discussion  in a revised version of our paper.
>
>
> $\textbf{[Diversity assumption.]}$ Indeed,  relaxing the diversity assumption (which is required in our paper only in the case of unknown context distribution) is an interesting direction for future work we had also identified. Since the central learner can broadcast parameters to the agents, the central learner can also coordinate the exploration. However, the main challenge that we faced in relaxing the diversity assumption for LinUCB-based algorithms is that the regret of LinUCB is bounded as $\tilde{O}(\sqrt{T}\\|\hat{\theta_T}-\theta_*\\|_{V_T})$.
>
> Without quantization, the quantity $\\|\hat{\theta_T}-\theta_*\\|_{V_T}$ grows slowly and is nearly a constant;
>
> however, without the diversity assumption, the quantization error can make $\\|\hat{\theta_T}-\theta_*\\|^2_{V_T}$ to grow as $\sqrt{T}$ in the worst case. This results in an extra factor of $T^{1/4}$ in the regret bound. The same problem persists if we use Thompson sampling. We are not sure if this is unavoidable or there are other analysis techniques that can result in better regret bounds.
>
> As other works that started with the diversity assumption, e.g., see [1-4] below, we hope this assumption can either be relaxed (for the case where the context distribution is unknown, perhaps assuming some partial knowledge of the distribution) in a future work, or if not to show that removing it will unavoidably increase the regret order.
> We are happy to include a discussion along the above lines in the revised paper.
>
> [1] I. Bogunovic, A. Losalka, A. Krause, and J. Scarlett, “Stochastic linear bandits robust to adversarial attacks,” in International Conference on Artificial Intelligence and Statistics. PMLR, 2021, pp. 991–999.
>
> [2] W. Wu, J. Yang, and C. Shen, “Stochastic linear contextual bandits with diverse contexts,” in International Conference on Artificial Intelligence and Statistics. PMLR, 2020, pp. 2392–2401.
>
> [3] M. Papini, A. Tirinzoni, M. Restelli, A. Lazaric, and M. Pirotta, “Leveraging good representations in linear contextual bandits,” in International Conference on Machine Learning. PMLR, 2021, pp. 8371–8380.
>
> [4] L. Li, Y. Lu, and D. Zhou, “Provably optimal algorithms for generalized linear contextual bandits,” in International Conference on Machine Learning. PMLR, 2017, pp. 2071–2080.

---

> ### Author Response · Authors · 2022-08-02
> **$\textbf{[Part 1]}$ Presentation comments**
>
> We thank the reviewer for the thorough review and for appreciating our contribution.
>
> $\textbf{[Comments to improve presentation.]}$ Thank you. We will incorporate the suggested changes above, as well as the response to the questions of the reviewer below in the next version.
>
> $\textbf{[Previous works require $O(d\log T)$ bits and exponential complexity.]}$ Previous works rely on constructing $1/T$-net  for the set of feature vectors. This requires $O(T^{d})$ vectors and the time complexity is also $O(T^d)$, which is exponential in d.
>
> $\textbf{[Distributed setting in [19]]}$ We agree [19] does not provide algorithms for distributed settings. However, they provide discretization schemes for actions (e.g., Lemma 20.1 [19]), and hence are related to our work.
>
> $\textbf{[Two notations for vector indices.]}$ We will correct this. The notation $A_i$ is shorter but it is confusing to use for expressions such as $(VX)_i$. We are happy to only use the second notation.
>
> $\textbf{[$\theta_*(\Lambda)$ notation.]}$ We wanted to emphasize that $\theta_*(\Lambda)$ is not necessarily the same as $\theta_*$. We will change this notation to $\theta_*'$ for clarification.
>
> $\textbf{[$\theta_t=\theta_*$ in line 220.]}$ This part was only meant to provide an intuition for the algorithm rather than a rigorous handling. We are happy to modify or remove this part in the next version of the paper.
>
> $\textbf{[Cited regret bound.]}$ Indeed the $\log$ inside the square root in the cited regret is a typo. We cite [19] for the regret bound of $O(d\sqrt{T}\log T)$ (see Corollary 19.3). We are aware that not all the regret bounds in [1,28,2] match the one in [19], however, as far as we know, they were among the first to introduce versions of LinUCB or Thompson sampling and analysis techniques that are the closest to [19].
> We will clarify this in the next version and include the papers that proved tighter bounds, thank you for pointing these out.
>
> $\textbf{[Conditioning on $\theta_*$.]}$ We wanted to put it in the same form as the term with $\hat{\theta_t}$, but the conditioning on $\theta_*$ can definitely be removed. Thanks for pointing out this. We will clarify it in the next version.

---

### Official Review · Reviewer_XFw3 · 2022-07-12

**Rating:** 7
**Confidence:** 4
**Soundness:** 4 excellent
**Presentation:** 3 good
**Contribution:** 3 good

**Summary:**

This paper studies contextual linear bandits with communication constraints, where it is desired to transmit as few number of bits as possible from the agent to the learner. The authors considered two different settings, i.e., the learner knows the distribution of contexts, and the learner doesn’t know the distribution of contexts, respectively. For the former case, the authors proposed an interesting reduction method that converts the original bandit problem with time-varying candidate set (multiple contexts) on the agent side to one with fixed candidate set (one context) on the learner side, so that there is no need to spend any bits on the contexts. For the latter case, the authors proposed an algorithm that quantizes the context with 5d bits via a stochastic quantizer. The authors proved that the proposed algorithms match the regret of standard contextual linear bandit algorithms.

**Questions:**

I am not sure if distributed contextual linear bandits a suitable title, as in this paper, learning only happens on the central learner, while the distributed agents are only responsible for pulling arms and compress the required messages. This is very different from the existing works in distributed contextual bandits, where each agent usually maintains its own model estimate.

The terms used for the agents in this problem should be made more consistent. Currently, we have "central agent", "central learner", and "learner" all referring to the same thing.

I am also a bit confused about the discussion about provacy in line 278. It is true that the proposed algorithms avoid directly sending context vectors, but isn't sending $\hat{\theta}$ also reveals private information, considering it will eventually converge to the true parameter $\theta_{\star}$.

**Strengths And Weaknesses:**

This paper is well written and easy to follow for the most part.
The proposed algorithm for the setting with known context distribution seems novel, and the technique may be useful for other related problems.
Recently there is an increasing number of works in distributed bandit learning that tries to minimize the total number of communication rounds needed in the time horizon. In comparison, this work aims to minimize the bits required for each round, so it may help further improve communication efficiency in a parallel direction.

My main concern is about the motivation for the current formulation.

As mentioned in introduction, the agents are assumed to be transient and "they may not be interested in learning - this may not be a task that the agents wish to consistently perform - and thus do not wish to devote resources to it". However, in the algorithm design, these agents still need to be cooperative, in the sense that they are required to perform computations to compress and communicate the required messages to help the learner, e.g. line 6-13, even though it will not benefit them at all.

Maybe I am missing something here. Why the focus is only put on the compression of context, while it is okay to transfer the uncompressed \hat{\theta}? For both algorithms, the learner still needs to send the uncompressed \hat{\theta} to the agent, which has the same dimension as the context vector for an arm, which is also communication expensive.

---

> ### Author Response · Authors · 2022-08-02
> **Cooperative agents assumption, unlimited downlink communication, title, privacy**
>
> We thank the reviewer for the thorough review and for appreciating our contribution.
>
> $\textbf{[Cooperative agents.]}$ Good question.
> We are considering a scenario where the user benefits from receiving an action (or policy, e.g., $\hat{\theta}$) from the central learner (e.g. a recommendation). In response, the user gives feedback to the central learner in terms of (compressed) context/reward. The compression operations benefit the user by helping reduce her communication cost. In principle, the user is not required to respond. But the central learner will be able to learn whenever there is a feedback. Creating an incentive for the user to respond is an interesting future topic.
> We are happy to further clarify this in the next version.
>
> $\textbf{[Downlink communication.]}$ Indeed the focus in this paper is on  algorithms that optimize the uplink communication (from the agents to the learner), and assume unlimited (cost-free) downlink communication. This is a standard assumption in wireless communication (e.g. see [1, 2, 3] below) for several reasons: uplink wireless links tend to be much more bandwidth restricted, since several users may be sharing the same channel as the central learner can be doing multiple tasks, and hence, may connect with multiple users simultaneously; uplink communication may also be battery-powered and thus more expensive to sustain; moreover, the agents may be less motivated to expend communication energy than the central learner who benefits from learning. Having said that,
> we note that we can quantize $\hat{\theta}$ without performance loss if the downlink were also communication constrained using $\approx 5d$ bits similar to quantizing $X_{t,a_t}$ in Algorithm 2. The analysis in the case of unknown context distribution is similar to the proof of Theorem 2.
>
> [1] M. H. Anisi, G. Abdul-Salaam, and A. H. Abdullah. A survey of wireless sensor network approaches and their energy consumption for monitoring farm fields in precision agriculture. Precision Agriculture, 16(2):216–238, 2015.
>
> [2] T. D. Novlan, H. S. Dhillon, and J. G. Andrews. Analytical modeling of uplink cellular networks. IEEE Transactions on Wireless Communications, 12(6):2669–2679, 2013.
>
> [3] O. A. Hanna, L. Yang, and C. Fragouli. Solving multi-arm bandit using a few bits of communication. In International Conference on Artificial Intelligence and Statistics, pages 11215–11236. PMLR, 2022.
>
>
> $\textbf{[Title.]}$ Thank you for the suggestion. We would be happy to change the title to “Learning from Remote Users in Contextual Linear Bandits Without Sharing the Context” or “Learning from Distributed Users in Contextual Linear Bandits Without Sharing the Context”.
>
> $\textbf{[Use only central learner.]}$ Thank you for pointing this. We are happy to change all of them to “central learner”.
>
> $\textbf{[Privacy.]}$ We do not make formal privacy claims in this paper, but simply observe that our approach could potentially be leveraged for privacy purposes. Having said that, we note that since $\theta_*$ is a property of the bandit model, not the user, it is not considered as private information, e.g., see [1,2,3] below. It is true that the reward can reveal some information about the context, e.g., if all the actions result in small reward for context and large reward for another context. However, privatizing the reward (which implies a private context in our case) is much easier than privatizing the context and there are many proposed optimal algorithms with little to no regret loss, e.g., see [4-7] below. This is not the case when privatizing the context. In fact it was shown in [1] that privatizing the context can lead to linear regret and relaxed definitions of privacy are proposed to avoid this.
>
> [1] R. Shariff and O. Sheffet. Differentially private contextual linear bandits. volume 31, 2018.
>
> [2] E. Garcelon, K. Chaudhuri, V. Perchet, and M. Pirotta. Privacy amplification via shuffling for linear contextual bandits. In International Conference on Algorithmic Learning Theory, pages 381–407. PMLR, 2022.
>
> [3] S. R. Chowdhury and X. Zhou. Shuffle private linear contextual bandits. arXiv preprint arXiv:2202.05567, 2022.
>
> [4] O. A. Hanna, A. M. Girgis, C. Fragouli, and S. Diggavi, “Differentially private stochastic linear bandits:(almost) for free,” arXiv preprint arXiv:2207.03445, 2022.
>
> [5] T. Sajed and O. Sheffet, “An optimal private stochastic-mab algorithm based on optimal private stopping rule,” in International Conference on Machine Learning. PMLR, 2019, pp. 5579–5588.
>
> [6] J. Tenenbaum, H. Kaplan, Y. Mansour, and U. Stemmer, “Differentially private multi-armed bandits in the shuffle model,” Advances in Neural Information Processing Systems, vol. 34, pp. 24 956–24 967, 2021.
>
> [7] W. Ren, X. Zhou, J. Liu, and N. B. Shroff, “Multi-armed bandits with local differential privacy,” arXiv preprint arXiv:2007.03121, 2020.

---

### Author Response · Authors · 2022-08-09
**Rebuttal revision updates**

We thank the reviewers for their valuable comments and time to review the paper. Please let us know if you have any additional concerns.

Based on the reviewers comments, we uploaded a revision of the paper with the following edits:
- Added a discussion on the cooperative agents (see line 81).
- Discussions on downlink communication are in line 47, line 364 and line 606.
- Updated the title to: "Learning from Distributed Users in Contextual Linear Bandits Without Sharing the Context".
- Added extra discussion on privacy (see line 583).
- Added extra discussion on the complexity of Algorithm 1 (computing $X^*(\theta)$) starting in line 591.
- Added a discussion on relaxing the diversity assumption starting in line 641.
- Explained how to break ties in the definition of $X^*(\theta)$ (see line 210).
- Added the observation that the boundedness assumption on $r_t$ can be relaxed (see line 131).
- Replaced "central agent" and "learner" with "central learner".
- Updated the cited regret bounds and included the references provided by the reviewer in line 126.
- Explained why previous algorithms use $O(d\log T)$ and what is meant by exponential complexity in line 66 and line 297.
- Removed conditioning on $\theta_*$ in the proof of Theorem 1 to avoid confusion.
- Changed $\theta_* (\Lambda)$ to $\theta_*'$.
- Removed $\theta_t=\theta_*$ in line 223 to avoid confusion and updated the explanation in that part.

---

### Meta-Review · Area_Chair_pb6b · 2022-08-25

**Recommendation:** Accept
**Confidence:** Certain

**Metareview:**

The reviewers are overall positive about the theoretical contributions of the paper, for which I share the same (generally) positive evaluation.
Please make sure you address all the reviewers' comments and incorporate them (and any new experimental results, if applicable) in your camera-ready.

**Award:**

No

---

### Decision · Program_Chairs · 2022-09-14

Accept